# Analysis of Corrected Graph Convolutions

**Robert Wang**[*]  **Aseem Baranwal** [†]  **Kimon Fontoulakis**[‡]

## Abstract

Machine learning for node classification on graphs is a prominent area driven by applications such as recommendation systems. State-of-the-art models often use multiple graph convolutions on the data, as empirical evidence suggests they can enhance performance. However, it has been shown empirically and theoretically, that too many graph convolutions can degrade performance significantly, a phenomenon known as oversmoothing. In this paper, we provide a rigorous theoretical analysis, based on the two-class contextual stochastic block model (CSBM), of the performance of vanilla graph convolution from which we remove the principal eigenvector to avoid oversmoothing. We perform a spectral analysis for $k$ rounds of corrected graph convolutions, and we provide results for partial and exact classification. For partial classification, we show that each round of convolution can reduce the misclassification error exponentially up to a saturation level, after which performance does not worsen. We also extend this analysis to the multi-class setting with features distributed according to a Gaussian mixture model. For exact classification, we show that the separability threshold can be improved exponentially up to $O(\log n/\log \log n)$ corrected convolutions.

## 1 Introduction

Graphs naturally represent complex relational information found in a plethora of applications such as social analysis [Backstrom and Leskovec, 2011], recommendation systems [Ying et al., 2018, Borisyuk et al., 2024], computer vision [Monti et al., 2017], materials science and chemistry [Reiser et al., 2022], statistical physics [Battaglia et al., 2016, Bapst et al., 2020], financial forensics [Zhang et al., 2017, Weber et al., 2019] and traffic prediction in Google Maps [Derrow-Pinion et al., 2021].

The abundance of relational information in combination with features of the corresponding entities has led to improved performance of machine learning models for classification and regression tasks. Central to the field of machine learning on graphs is the graph convolution operation. It has been shown empirically [Defferrard et al., 2016, Kipf and Welling, 2017, Gasteiger et al., 2019, Rossi et al., 2020] that using graph convolutions to the feature data enhances the prediction performance of a model, but too many graph convolutions can have the opposite effect [Oono and Suzuki, 2020, Chen et al., 2020, Keriven, 2022, Wu et al., 2023], an issue known as oversmoothing. Several solutions have been proposed for this problem, we refer the reader to the survey of Rusch et al. [2023].

In this paper we provide a rigorous spectral analysis, based on the contextual stochastic block model [Deshpande et al., 2018], to show that the oversmoothing phenomenon can be alleviated by excluding the principal eigenvector's component from the graph convolution matrix. This is similar to the state-of-the-art normalization approach used in Zhao and Akoglu [2020], Rusch et al. [2023] (PairNorm). However, our method explicitly uses the principal eigenvector. We provide below some intuition about why excluding the principal eigenvector helps to alleviate over-smoothing.

---

[*]Cheriton School of Computer Science, University of Waterloo
[†]Cheriton School of Computer Science, University of Waterloo
[‡]Cheriton School of Computer Science, University of Waterloo

38th Conference on Neural Information Processing Systems (NeurIPS 2024).

Let $A$ be the adjacency matrix of the given graph, and $D$ be the degree matrix. Vanilla graph convolutions are represented using matrices such as $D^{-1}A$ or $D^{-1/2}AD^{-1/2}$ [Kipf and Welling, 2017]. Suppose our graph is $d-$regular, meaning that each node has exactly $d$ neighbors. In this case, both graph convolutions reduce to $\frac{1}{d}A$. The top eigenvector of $A$ is $\mathbb{1}$ with eigenvalue $d$, where $\mathbb{1}$ is the vector of all ones. This means that $\lim_{k\to\infty}\frac{1}{d^k}A^k = \frac{1}{n}\mathbb{1}\mathbb{1}^\top$, which implies that applying many convolutions is equivalent to projecting our data onto the all-ones vector. Thus, all feature values will converge to the same point. Therefore, we should expect, as verified by most real-world and synthetic experiments, that many rounds of the convolution $x \mapsto \frac{1}{d}Ax$ will lead to a large learning error. However, if we instead perform convolution with the corrected matrix $\tilde{A} := \frac{1}{d}A - \frac{1}{n}\mathbb{1}\mathbb{1}^\top$, then the convergent behavior of $x \mapsto \tilde{A}^k x$ would be equivalent to projecting $x$ onto the *second* eigenvector of $A$. This eigenvector is known to capture information about sparse bipartitions in the graph $G$ [Cheeger, 1970, Alon and Milman, 1985, Alon, 1986], and so for certain problems, like binary classification, we may expect this eigenvector to capture a larger amount of information about our signal. We note that another well-studied graph matrix is the Laplacian, D-A. In the regular case, this has the same eigenvectors as the adjacency matrix, but with reversed spectrum. The trivial eigenvector we remove is exactly the Nullspace of the Laplacian.

In our analysis, we study the classification problem in the contextual stochastic block model, with a focus on linear, binary classification. Our results are stated in terms of the following corrected convolution matrices:

$$\hat{A} = D^{-1/2}AD^{-1/2} - \frac{1}{\mathbb{1}^\top D\mathbb{1}}D^{1/2}\mathbb{1}\mathbb{1}^\top D^{1/2} \qquad \text{and} \qquad \tilde{A} = \frac{1}{d}A - \frac{1}{n}\mathbb{1}\mathbb{1}^\top, \qquad (1)$$

where $d := 2|E|/n$ is the empirical average degree in $A$, where $|E|$ is the number of edges in the graph. Note that $\hat{A}$ is derived from the *normalized* adjacency matrix, while $\tilde{A}$ is (up to a scalar multiple) its *unnormalized* counterpart. Briefly, we demonstrate that when the graph is of reasonable quality, the corrected graph convolutions exponentially improve both partial and exact classification guarantees. Depending on the density and quality of the given graph, improvement becomes saturated after $\mathcal{O}(\log n)$ convolutions in our partial and exact classification results. However, in comparison to a similar analysis in [Wu et al., 2023] for vanilla graph convolutions (without correction), we show that classification accuracy does not become worse as the number of convolutions increases.

## 1.1 Our Contributions

In this work, we provide, to our knowledge, the first theoretical guarantees on partial and exact classification after $k$ rounds of graph convolutions in the contextual stochastic block model. Our main result is to show that each graph convolution with the corrected matrix reduces the classification error by a multiplicative factor until a certain point of "saturation" and the number of convolutions required until saturation depends on the amount of input feature variance. We show that the accuracy of the linear classifier at the point of saturation only depends on the strength of the signal from the graph. This is in contrast to the uncorrected convolution matrix, which will always exhibit a decrease in classification accuracy after many convolutions. Finally, we show that given slightly stronger assumptions on graph density and signal strength, the convolved data at the point of saturation will be linearly separable with high probability. To quantify our results, we let $p$ and $q$ be the intra- and inter-class edge probabilities with $\gamma(p, q) = |p - q|/(p + q)$ being the "relative signal strength" in the graph. Let $\bar{d}$ be the expected degree of each vertex. Our results can be summarized as follows:

- If $p + q \geq \Omega(\frac{\log^2 n}{n})$ and $\gamma \geq \Omega(\frac{1}{\sqrt{\bar{d}}})$, each convolution with $\tilde{A}$ reduces classification error by a factor of about $\frac{1}{\gamma^2 d}$ until the fraction of errors is $O(\frac{1}{\gamma^2 d})$

- If $p + q \geq \Omega(\frac{\log^2 n}{n})$ and $\gamma \geq \Omega(\sqrt{\frac{\log n}{d}})$, each convolution with $\hat{A}$ reduces classification error by a factor of about $\frac{\log n}{\gamma^2 d}$ until the fraction of errors is $O(\frac{\log n}{\gamma^2 d})$

- If $p + q \geq \Omega(\frac{\log^3 n}{n})$, $\gamma \geq \Omega(k\sqrt{\frac{\log n}{d}})$ and the input features has signal-to-noise ratio at least $\Omega(\sqrt{\frac{\log n}{n}})$, the data is linearly separable after $k$ rounds of convolutions with $\tilde{A}$.

To obtain our partial classification results, we use spectral analysis to bound the mean-squared-error between the convolved features and the true signal. For exact classification, we prove a concentration

inequality on the total amount of message received by a vertex through "incorrect paths" of length $k$ after $k$ rounds of convolution through a combinatorial moment analysis. Using this, we establish entry-wise bounds on the deviation of the convolved feature vector from the true signal.

Finally, we extend our partial-recovery result to the multi-class setting. In this setting, we assume our features are distributed according to a Gaussian mixture model with $L$ equal-sized clusters and our graph is distributed according to a $L$-block stochastic block model. Our analysis for partial recovery generalizes easily to the multi-class setting with the use of basic non-linear classifiers. Just as before, we show that convolution with the corrected, un-normalized adjacency matrix, $\tilde{A}$, reduces classification error by a constant fraction each round, until a point of saturation where no further improvement is made.

## 2 Literature review

Research on graph learning has increasingly focused on methods that integrate node features with relational information, particularly within the semi-supervised node classification framework, see, for example, Scarselli et al. [2009], Cheng et al. [2011], Gilbert et al. [2012], Dang and Viennet [2012], Günnemann et al. [2013], Yang et al. [2013], Hamilton et al. [2017], Jin et al. [2019], Mehta et al. [2019], Chien et al. [2022], Yan et al. [2021]. These studies have underscored the empirical advantages of incorporating graph structures when available.

The literature also addresses the expressive capacity [Lu et al., 2017, Balcilar et al., 2021] and generalization potential [Maskey et al., 2022] of Graph Neural Networks (GNNs), including challenges like oversmoothing [Keriven, 2022, Xu et al., 2021, Oono and Suzuki, 2020, Li et al., 2018, Rusch et al., 2023]. In our paper, we ground our work on the contextual stochastic block model [Deshpande et al., 2018], a widely used statistical framework for analyzing graph learning and inference problems. Recent theoretical studies have extensively used the CSBM to illustrate several statistical, information-theoretic, and combinatorial results on relational data accompanied by node features. In Deshpande et al. [2018], Lu and Sen [2020], the authors investigate the classification thresholds for accurately classifying a significant portion of nodes from this model, given linear sample complexity and large but bounded degree. Additionally, Hou et al. [2020] introduces graph smoothness metrics to quantify the utility of graphical information. Further developments in Chien et al. [2021, 2022], Baranwal et al. [2021, 2023a] extend the application of CSBM, establishing exact classification thresholds for graph convolutions in multi-layer networks, accompanied by generalization guarantees. A theoretical exploration of the graph attention mechanism (GAT) is provided by Fountoulakis et al. [2023], delineating conditions under which attention can improve node classification tasks. More recently, Baranwal et al. [2023b] provide the locally Bayes optimal message-passing architecture for node classification for the general CSBM.

In this paper, we provide exact and partial classification guarantees for multiple graph convolution operations. Previous investigations have often been confined to a few convolution layers, limiting the understanding of their effects on variance reduction (see, for example, Baranwal et al. [2023a]). Our findings contribute a novel spectral perspective on graph convolutions, describing how the fraction of recoverable nodes is influenced by the signal-to-noise ratio in the node features and the scaled difference between intra- and inter-class edge probabilities. We also demonstrate that the oversmoothing phenomenon can be alleviated by excluding the principal eigenvector's component from the adjacency matrix – a strategy somewhat akin to the normalization approach used in Zhao and Akoglu [2020] (PairNorm), albeit our method explicitly uses the principal eigenvector and is grounded in rigorous spectral justifications.

In a relatively recent work [Wu et al., 2023] the authors rigorously analyze the phenomenon of oversmoothing in GNNs for the 2-block CSBM by identifying two competing effects of graph convolutions: the mixing effect, which homogenizes node representations across different classes, and the denoising effect, which homogenizes node representations within the same class. Their analysis shows that oversmoothing occurs when the mixing effect dominates the denoising effect, and they quantify the number of layers required for this transition. In contrast, we work with the corrected graph convolution in the 2-block CSBM and show that it improves performance exponentially up to saturation, after which more convolutions do not improve nor degrade performance. On a technical level, the previous work only analyzes the distribution of a single node's feature values after convolution and does not take into account correlations between nodes. In our work, we use spectral

analysis of higher powers of the convolution matrix, which takes into account correlations between nodes to obtain our partial classification results over the whole dataset. To handle the modified convolution in the exact classification setting, we analyze the error more directly through matrix perturbation analysis rather than trying to directly count the higher-order neighbors of each vertex as in previous works [Baranwal et al., 2023a, Wu et al., 2023].

# 3 Preliminaries and Model Description

Throughout this paper, we use $\mathbb{1}$ to denote the all-ones vector and $e_i$ to denote the $i^{th}$ standard basis vector in $\mathbb{R}^n$. Given a vector $x \in \mathbb{R}^n$, we use $\|x\|$ to denote its Euclidean norm $\sqrt{\sum_{i=1}^n x(i)^2}$. We use $\|x\|_\infty$ to denote its infinity norm, $\max_{i=1}^n |x(i)|$. For a matrix $M \in \mathbb{R}^n$, we use $\|M\|$ to denote its operator norm, $\max_{x \neq 0, \|x\|=1} \|Mx\|$. We use $\|M\|_F = \sqrt{\sum_{i,j} M_{i,j}^2}$ to denote its Frobenius norm. We also make routine use of the spectral theorem, which says that if $M$ is a $n \times n$ symmetric matrix, then it can be diagonalized with $n$ orthogonal eigenvectors and real eigenvalues. In particular, there exist $\lambda_1, \lambda_2, ... \lambda_n \in \mathbb{R}$ and orthonormal vectors $w_1, w_2, ... w_n \in \mathbb{R}^n$ such that $M = \sum_{i=1}^n \lambda_i w_i w_i^\top$. Note that when $M$ is symmetric, $\|M\| = \max_i |\lambda_i| = \max_{x:\|x\|=1} |x^\top M x|$.

Finally, we use the $\mathcal{N}(\mu, \Sigma)$ to a Gaussian distribution with mean $\mu$ and covariance matrix $\Sigma$. For one-dimensional Gaussians, we use $\mathcal{N}(\mu, \sigma^2)$. For $X \sim \mathcal{N}(\mu, \sigma^2)$, we will frequently use the Gaussian tail bound: $\mathbf{Pr}\left[|X - \mu| > t\sigma\right] \leq \exp(-\frac{t^2}{2})$.

## 3.1 Contextual Stochastic Block Model

In this section, we formally describe the contextual stochastic block model introduced by [Deshpande et al., 2018]. Our model is defined by parameters $n, m \in \mathbb{N}$, $p, q \in [0, 1]$, $\mu, \nu, \in \mathbb{R}^m$ and $\sigma \in \mathbb{R}^+$ In the model, we are given a random undirected graph, $G = (V, E)$, where $|V| = n$, drawn from the 2-block stochastic block model and features drawn from the Gaussian mixture model. Our vertices are partitioned into two classes, $S$ and $T$, of size $n/2$, which we want to recover. For each pair of vertices $i, j \in S$ and $i, j \in T$, the edge $(i, j)$ is in $E$ independently with probability $p$ while for each pair $i \in S$ and $j \in T$, the edge $(i, j)$ is in $E$ with probability $q$. In addition to the graph, we are also given a feature matrix $X \in \mathbb{R}^{n \times m}$ drawn from a Gaussian mixture model with two centers $\mu$ and $\nu$. For each $i \in V$, we let $g_i \sim \mathcal{N}(0, \sigma^2 I_m)$ be an i.i.d. Gaussian noise vector. Now let $(x_i)_{i \in n}$ be the rows of $X$. For $i \in S$, we have $x_i = \mu + g_i$ and for each $i \in T$, we have $x_i = \nu + g_i$.

In the multi-class setting, our nodes are partitioned into $L$ classes, $\mathcal{C}_1, ... \mathcal{C}_L$, of size $n/L$. The inter-class edge probability is $p$ and intra-class edge probability is $q$. We assume the features are generated by a Gaussian mixture with $L$ centers $c_1, ... c_L \in \mathbb{R}^m$. If node $i$ is in class $l$, then we observe its feature vector as $x_j = c_l + g_i$. In addition, we will let $\mu_i := c_l$, for $i \in \mathcal{C}_l$, denote the center for vertex $i$.

# 4 Results and Interpretation

In our analysis, there are two types of objectives. In the exact classification objective, the aim is to exactly recover $S$ and $T$ with probability $1 - o(1)$. In the partial classification, or "detection" problem, the goal is to correctly classify $1 - o(1)$ fraction of vertices correctly with probability $1 - o(1)$. We begin by stating our results for the partial classification regime. For ease of notation, we will assume $p > q$ from this point forward. We show in Appendix B.1 that this assumption is made without loss of generality.

**Theorem 4.1.** *Suppose we are given a 2-block $m$-dimensional CSBM with parameters $n, p > q, \mu, \nu, \sigma$ satisfying $\gamma(p, q) := \frac{p-q}{p+q} \geq \Omega\left(\sqrt{\frac{1}{np}}\right)$ and $p \geq \Omega\left(\frac{\log^2 n}{n}\right)$. There exists a linear classifier such that after $k$ rounds of convolution with $\tilde{A}$, will, with probability at least $1 - \frac{1}{2}\exp(-\Omega\left(\frac{n\|\mu-\nu\|^2}{\sigma^2}\right))$, misclassify at most*

$$O\left(\frac{1}{\gamma^2 p} + \left(\frac{C}{\gamma\sqrt{np}}\right)^{2k} \frac{\sigma^2}{\|\mu - \nu\|^2} n \log n\right)$$

*vertices, where $C$ is an absolute constant. Furthermore, if $\gamma \geq \Omega(\sqrt{\frac{\log n}{np}})$ then with probability at least $1 - \frac{1}{2}\exp(-\Omega(\frac{n\|\mu-\nu\|^2}{\sigma^2}))$, the same linear classifier after $k$ rounds of convolution with $\hat{A}$ will misclassify at most*

$$O\left(\frac{\log n}{\gamma^2 p} + \left(\frac{C\log n}{\gamma\sqrt{np}}\right)^{2k}\frac{\sigma^2}{\|\mu-\nu\|^2}n\log n\right)$$

*vertices.*

Now we take a closer look at the error bound. For brevity, we will focus on our results regarding convolutions with $\tilde{A}$. First, we see that an important ratio in our bound is the term $1/(\gamma^2 np)$. This term is small if $\gamma^2$ is much larger than the inverse of the expected degree of each vertex, $(p+q)n/2$, which is at most $np$. Our assumption that this ratio is upper bounded by a constant means that we need the signal from the graph to be sufficiently strong. Now, if we examine our misclassification error bound, and let $\rho = C/(\gamma^2 np)$ where $C$ is a sufficiently large constant, then we see that the *fraction* of misclassified vertices is at most $\rho + \rho^k \sigma^2 \log n/\|\mu-\nu\|^2$. Our assumption on the parameters ensures that $\rho < 1$. Note that only the second term depends on $k$, and the feature's noise-to-signal ratio. This term measures the amount of error introduced by the variance in the features and exponentially decreases with $k$. Moreover, after about $k = \log_{1/\rho}\left(\sigma^2 \log n/(\rho\|\mu-\nu\|^2)\right)$ convolutions, the $\rho$ term, which only depends on graph parameters, will dominate over the variance term, indicating that more convolutions will not improve the quality of the convolved features beyond the quality of the signal from the graph. If $\sigma/\|\mu-\nu\|$ is constant, we will always reach our optimal error bound of $O(\rho)$ when $k = O(\log\log n)$. Moreover, if $\gamma = \Omega(1)$, as was assumed in Baranwal et al. [2021], then we will have $1/\rho \geq \Omega(np)$. This means even when $\sigma/\|\mu-\nu\| \approx \sqrt{n/\log n}$, we will reach optimality in constant number of convolutions with high probability if the graph is moderately dense. For example, if $p = 1/\sqrt{n}$, then we only need 3 convolutions and if $p = \Omega(1)$, then we only need 2. On the other hand, if $\gamma$ is on the order of $\Theta(1/\sqrt{np})$, then in the worst case, we may need $\log n$ convolutions to reach our optimal bound. Next, we state our results for exact classification.

**Theorem 4.2.** *Suppose we are given a 2-block $m$-dimensional CSBM with parameters $n, p > q, \mu, \nu, \sigma$ satisfying $\gamma(p,q) \geq \Omega\left(k\sqrt{\frac{\log n}{np}}\right)$ and $p \geq \frac{\log^3 n}{n}$. Then after $k = O(\log n)$ rounds of graph convolution with $\tilde{A}$, our data is linearly separable with probability $1 - n^{-\Omega(1)}$ if*

$$\frac{\|\mu-\nu\|}{\sigma} \geq \Omega\left(\max\left(\sqrt{\frac{\log n}{n}}, \left(\frac{C}{\gamma\sqrt{np}}\right)^k\sqrt{\log n}\right)\right)$$

*where $C$ is an absolute constant.*

Here, we bound the minimum signal-to-noise ratio required for exact classification as a function of $p, q, n$ and $k$. Just like the partial classification result, our function has a term that decreases exponentially with $k$ and a term independent of $k$. The rate of decrease for the dependent term is proportional to $1/(\gamma\sqrt{np})$, or $\sqrt{\rho}$. We see once again that with more convolutions, the requirement on the feature signal-to-noise ratio for exact classification becomes exponentially weaker. More-over, since we assumed that $\gamma \geq \Omega(k\sqrt{\log n/(np)})$, as long as $\|\mu-\nu\| \geq \Omega(\sigma\sqrt{\log n/n})$, the data becomes linearly separable after $k = O(\log n/\log\log n)$ convolutions. Just as in the partial classification case, we observe that the larger $\gamma$ is, the fewer convolutions we need to obtain the optimal bound. In particular, if $\gamma = \Omega(1)$ and $p = \Omega(1)$ then one convolution already gives the optimal bound, and if $p = 1/\sqrt{n}$, then two convolutions are enough. For technical reasons, we only analyze exact classification using convolution with the corrected un-normalized adjacency matrix, $\tilde{A}$. Similar bounds should hold for $\hat{A}$ based on our simulation results, but we leave this for future work.

## 4.1 Discussion on our Assumptions

Both Theorem 4.1 and Theorem 4.2 require a lower bound of $\gamma \geq \omega(1/\sqrt{np})$, and this is to ensure that the signal from the graph is strong enough so that a convolution does not destroy the signal from the data. Also, implicit in the probability bound of Theorem 4.1 and in Theorem 4.2, is the assumption that our signal-to-noise ratio, $\|\mu-\nu\|/\sigma$, is at least $\omega(1/\sqrt{n})$ so the feature noise does

not completely drown out the signal. Our lower-bound assumption on $p$ is to ensure concentration in the behavior of the degrees and the adjacency matrix towards their expectation. In Theorem 4.2, we also assume that $k = O(\log n)$. This is done mainly for technical reasons of our proof but we note that this assumption is made without loss of generality because as mentioned, the bound in Theorem 4.2 does not improve for $k \geq \log n$. Finally, we note that the case $p > q$ corresponds to a homophilous graph, and the case $p < q$ corresponds to a heterophilous graph (see Luan et al. [2021], Ma et al. [2022] for more). For binary classification, it has been shown [Baranwal et al., 2023a] that one can assume $p > q$ without loss of generality and make corresponding adjustments in the classifier. As such, we assume that $p > q$. For more detail regarding this assumption, see Appendix B.1.

## 5    One-Dimensional CSBM

In Baranwal et al. [2021], the authors showed that analyzing the linear classifier for the $m$-dimensional CSBM reduces to analyzing the 1-dimensional model. We say that a CSBM is **one-dimensional and centered** with parameters $n, p, q, \sigma$ if it has one-dimensional features and means $1/\sqrt{n}$ and $-1/\sqrt{n}$. That is, we have one feature vector $x \in \mathbb{R}^n$ given by $x = s + g$, where $g \sim \mathcal{N}(0, \sigma^2 I_n)$ and $s(i) = 1/\sqrt{n}$ for $i \in S$ and $-1/\sqrt{n}$ for $i \in T$. We will refer to $s$ as our *signal* vector and for ease of notation, we normalize it so that it always has unit norm. The following lemma allows us to reduce the analysis of the linear classifier for a general CSBM to the analysis of the 1-dimensional centered CSBM (proof in Appendix B.1). Thus, in the proofs of our main theorems, we will analyze the 1-dimensional case before applying Lemma 5.1.

**Lemma 5.1.** *Given an $m$-dimensional 2-block $CSBM$, there exist $w \in \mathbb{R}^m$ and $b \in \mathbb{R}^n$ such that $Xw + b = s + g$ where for each vertex $i$, $g_i$ is i.i.d. $\mathcal{N}(0, \sigma'^2)$ with $\sigma' = \frac{4\sigma}{\sqrt{n}\|\mu - \nu\|}$.*

In the 1-dimensional model, it is clear how our signal $s$ is present in our features. Our convolution matrix also captures the signal because it can be viewed as a perturbation of the matrix $ss^\top$. This is especially evident with the un-normalized convolution matrix $\tilde{A}$, which satisfies the following

$$\tilde{A} = \eta ss^\top + \frac{1}{d}R + d'\mathbb{1}\mathbb{1}^\top \tag{2}$$

where $\eta := (p - q)n/(2d)$ is the *signal strength*, $d' := ((p + q)n/2 - d)/(nd)$ is the *average* degree deviation, and $R$ is the "edge-deviation" matrix, where $R_{i,j} = A_{i,j} - \mathbb{E}[A_{i,j}]$. Since $R$ has i.i.d. zero-mean entries with variance at most $p$, we can use standard matrix concentration inequalities to show it is not too big. Likewise, $d'$ is small due to degree concentration, and together, these two concentration results imply that $\tilde{A}$ is close to $\gamma ss^\top$. In fact, if we show degree concentration for all vertices, then we can show that $\hat{A}$ also behaves like $\tilde{A}$. We state these concentration results below.

**Proposition 5.2.** *Assume that $p = \Omega(\frac{\log^2 n}{n})$, and let $\gamma = \frac{p-q}{p+q}$. With probability $1 - n^{-\Omega(1)}$, we have the following concentration results*

1. *$|d'| \leq O(1/n^{1.5})$, which implies that $\eta \in \gamma(1 \pm o(1))$.*

2. *$\|R\| \leq O(\sqrt{np})$*

3. *$\left\|\tilde{A} - \gamma ss^\top\right\| \leq O(\frac{1}{\sqrt{np}})$ and $\left\|\hat{A} - \gamma ss^\top\right\| \leq O(\sqrt{\frac{\log n}{np}})$*

These concentration properties are crucial for spectral analysis. Details are given in Appendix B.1.

## 6    Partial Classification

In this section, we give a sketch of the proof of our partial classification result, Theorem 4.1. The full proofs can be found in Appendix C. We will show that partial classification can be achieved if the convolved vector is well-correlated with our signal vector $s$ as defined in the beginning of Section 5 in the centered-1 dimensional CSBM. We will analyze our result for convolutions using the matrix $M \in \{\tilde{A}, \hat{A}\}$.

**Proposition 6.1.** *Given a centered 1-dimensional 2 block CSBM with parameters $n, p > q$, $\sigma$ and $\gamma(p, q) = \frac{p-q}{p+q}$, suppose our convolution matrix $M$ satisfies $\left\|M - \gamma ss^\top\right\| \leq \delta$ and $\gamma \geq C\delta$ for a*

*large enough constant C. Let $x_k = M^k x$, be the result of applying k rounds of graph convolution to our input feature x. Then with probability at least $1 - \frac{1}{2}\exp(-\frac{1}{4\sigma^2})$, there exists a scalar $C_k$ and an absolute constant $C'$ such that*

$$\|C_k x_k - s\|^2 \le O\left(\frac{\delta^2}{\gamma^2} + \left(\frac{C'\delta}{\gamma}\right)^{2k} n\sigma^2 \log n\right)$$

The main idea of our analysis is to use the fact that the top eigenvector of $M$, denoted $\hat{s}$, is well correlated with our signal $s$. Since $\|M - \gamma ss^\top\| \le \delta$ and $\gamma > C\delta$ by assumption, standard Matrix perturbation arguments imply that the spectrum of $M$ will be in $(\gamma \pm \delta, \pm\delta, \pm\delta, ...)$ with high probability. Given our assumption of $\gamma > C\delta$, there will be a large gap between the top eigenvalue of $M$ and the rest of its eigenvalues. A well-known result Davis and Kahan implies that $\|s - \hat{s}\|^2 \le O(\delta^2/\gamma^2)$, i.e. $s$ is close to $\hat{s}$. Thus, we prove Proposition 6.1 by showing that the influence of the rest of the eigenvectors on our convolved vector, $x_k = M^k x$, decreases exponentially with $k$, which allows us to bound the squared norm distance between $x_k$ and $s$. In particular, we take our normalization constant to be $C_k \approx 1/\lambda_1^k$, where $\lambda_1$ is the maximum eigenvalue of $M$. Note that $\lim_{k\to\infty}(1/\lambda_1^k)M^k = \hat{s}\hat{s}^\top$. Roughly speaking, we decompose our convolution vector as $C_k x_k \approx (1/\lambda_1^k)M^k s + (1/\lambda_1^k)M^k g$. To bound the distance of this vector from $s$, we analyze the contribution of each of the two terms to our error separately. That is, we show that with high probability $\|(1/\lambda_1^k)M^k s - s\|^2 \le O(\delta^2/\gamma^2)$ and $\|M^k g\|^2 \le O((\delta/\gamma)^{2k} n\sigma^2 \log n)$. Note that the first error term is from taking the convolution of the noisy graph with the true signal, and thus does not decrease with $k$. The second error term, on the other hand, comes from variance in the features, $g$, and thus decreases with our noise level $\sigma$ and drops exponentially with each convolution.

Finally, given Proposition 6.1, we can prove the partial classification result by noting that if we partition the convolved 1-dimensional data around 0, then each misclassified vertex contributes $1/n$ to the mean-squared error, which means the number of misclassified vertices is at most $\|C_k x_k - s\|^2 n$. This, combined with Lemma 5.1 to generalize to the $m-$dimensional case will prove Theorem 4.1

## 7  Exact Classification

In this section, we sketch the proof of Theorem 4.2 for exact classification using the un-normalized corrected convolution matrix $\tilde{A}$. Full proofs can be found in Appendix D. To show linear separability, we would like $x_k = \tilde{A}^k x$ to have positive entries for all vertices in $S$ and negative entries for all vertices in $T$. This means that we want to show $\|C_k x_k - s\|_\infty < 1/\sqrt{n}$ for some appropriate scalar $C_k$. In particular, we will take $C_k$ to be $1/\eta^k$, where $\eta$ is our empirical estimate of $\gamma(p, q)$. In partial classification, it sufficed to bound the mean squared error $\|C_k x_k - s\|_2^2$, using spectral analysis but bounding $\|C_k x_k - s\|_\infty$ requires more work because now we are bounding the *entrywise* instead of average error. In our approach, we bound the volume of messages passed through "incorrect paths" in our graph and show that the contribution from these messages is small. Then, we show the other source of error, the feature variance, is reduced exponentially with each round of convolution. As with the partial classification result, we first prove our result in the 1-dimensional centered model:

**Proposition 7.1.** *Suppose we are given a 1-dimensional centered 2-block CSBM with parameters $n, p, q, \sigma$ and $k = O(\log n)$ such that $\gamma = \frac{p-q}{p+q} \ge \Omega\left(k\sqrt{\frac{\log n}{np}}\right)$, $p \ge \frac{\log^3 n}{n}$, and $\sigma \le O\left(\frac{1}{\sqrt{\log n}}\right)$. Then with probability at least $1 - n^{-\Omega(1)}$:*

$$\left\|\frac{1}{\eta^k}\tilde{A}^k x - s\right\|_\infty \le \frac{1}{2\sqrt{n}} + O\left(\left(\frac{C}{\gamma\sqrt{np}}\right)^k \sigma\sqrt{\log n}\right)$$

Given, Proposition 7.1, Theorem 4.2 follows immediately by applying Lemma 5.1. We now give a sketch of our proof for Proposition 7.1. To bound each entry of $C_k x_k - s$, we must bound $|e_u^\top \tilde{A}^k x/\eta^k - e_u^\top s|$ for all $u \in V$. Similar to in partial classification, we will split our error into error from $\tilde{A}^k s$ and error from $\tilde{A}^k g$. That is, for each $u \in V$, we separately upper bound $|e_u^\top(\eta ss^\top + R')^k s - e_u^\top s|$ and $|e_u^\top(\eta ss^\top + R')^k g|$, where $R' = \tilde{A} - \eta ss^\top$. The matrix $(\eta ss^\top + R')^k$, when expanded out, can be written as $\eta^k ss^\top$ plus a sum of $2^k - 1$ terms, each of which is a non-commutative product of matrices of the form $\eta ss^\top$ or $R'$. We group these error matrices into terms

of order $\ell$ for $\ell \in [k]$, where the $\ell^{th}$ order terms are comprised of products that contain $\ell$ copies of $R'$ and $k - \ell$ copies of $\gamma ss^\top$.

In our analysis, we first use degree concentration to show that instead of analyzing the error w.r.t. $R'$, it suffices to analyze the error w.r.t. $\frac{1}{d}R$. To bound $e_u^\top(\eta ss^\top + \frac{1}{d}R)^k s$, we expand it out and find that each term arising from an error matrix of order $\ell$ can be written as a multiple of $e_u^\top R^{a_1} s \cdot s^\top R^{a_2} s \cdot ... s^\top R^{a_L} s$ where $a_1, ... a_L$ are non-negative integers satisfying $a_1 + a_2 + ... a_L = \ell$. The symmetric terms can be bounded by showing $s^\top R^a s \lesssim (\sqrt{np})^a$ using simple spectral arguments. To control the asymmetric term, we show that with high probability, $|e_u^\top R^{a_1} s| \leq \frac{1}{\sqrt{n}}(Cnp \log n)^{a_1/2}$ for a constant $C$. This part is the most technical part and requires the slightly stronger graph density assumption: $p \geq \Omega(\log^3 n/n)$. Thus, we have $|e_u^\top R^{a_1} s \cdot s^\top R^{a_2} s \cdot ... s^\top R^{a_L} s| \leq \frac{1}{\sqrt{n}}(Cn \log n)^{\ell/2}$ for some constant $C$. The analysis for bounding $e_u \tilde{A}^k g$ is similar. Finally, by combining these bounds with our assumptions that $\gamma$ is large enough, we obtain Proposition 7.1.

Now, we take a closer look at the step of bounding the asymmetric term. Recall that $R$ is a random symmetric matrix with i.i.d. zero mean entries. The term $e_u^\top R^\ell s$ can be expressed as $\sum_w R_{w(0),w(1)} R_{w(1),w(2)}, ... R_{w(\ell-1),w(a)} s(w(a))$ where the sum is over walks, $w$, of length $a$ in the complete graph over $n$ vertices starting at $w(0) := u$. From a message-passing perspective, one can interpret this as bounding the deviation between the amount of signal message $u$ receives over paths of a certain length and the amount of message it *expects* to receive. To bound this term, we use a path counting argument to control its higher moments, and then apply Markov's inequality: $\mathbf{Pr}\left[|e_u^\top R^a s| \geq \lambda\right] < \frac{1}{\lambda^{2t}}\mathbb{E}[|e_u^\top R^a s|^{2t}]$.

# 8 Multi-class Analysis on Gaussian Mixture Model

In this section, we will formally state and sketch our results for the multi-class analysis. Full proofs are in Appendix E. For simplicity, we will only analyze convolution with the un-normalized corrected convolution matrix, $\tilde{A}$. The reason that the corrected convolution still gives good performance is that when class sizes are balanced, the second eigenspace of the *expected* adjacency matrix has multiplicity $L - 1$ and exactly captures the $L$ clusters (see Lemma E.1). Before formally stating our result, we will introduce some useful notation:

- **Graph Signal:** $\lambda := \frac{(p-q)n}{dL}$ is the strength of the signal from the graph.
- **Graph Noise:** $\delta := C(\frac{1}{d}(\sqrt{np(1-p)/L} + \sqrt{nq(1-q)}))$ for some constant $C$. $\delta$ is an upper bound on the graph noise.
- Let $U := \mathbb{E}[X]$ be the matrix whose $i^{th}$ column is $\mu_i$. We also assume our features are centered on expectation so that $U^\top \mathbf{1} = 0$. This is not restrictive since it can be satisfied by applying a linear shift to the features
- Let $\Delta = \min_{i,j \in [n]} \|\mu_i - \mu_j\|$ be the minimum distance between the centers

**Theorem 8.1.** *Given the CSBM with parameters, $p, q, L, n, m$, suppose $\min(p,q) \geq \Omega(\frac{\log^2 n}{n})$ and $|\lambda| > 4k\delta$. Let $X^{(k)} = \frac{1}{\lambda^k}\tilde{A}^k X$ be the feature matrix after $k$ rounds of convolutions with scaling factor $1/\lambda^k$. Let $x_i^{(k)}$ be the $i^{th}$ row of the matrix $X^{(k)}$. Then with probability $1 - n^{-\Omega(1)}$, at least $n - n_e$ nodes, $i$, satisfy $\left\|x_i^{(k)} - \mu_i\right\| < \Delta/2$ where*

$$n_e = O\Big((k\delta/|\lambda|)^2 \frac{\|U\|_F^2}{\Delta^2} + (L + n(\delta/|\lambda|)^{2k})\frac{\sigma^2 m \log n}{\Delta^2}\Big).$$

*In particular, the quadratic classifer $x \mapsto softmax(\|x - c_l\|^2)_{l=1}^L$ will correctly classify at least $n - n_e$ points, and when $n_e = o(n)$, then we can correctly classify $1 - o(1)$ fraction of points.*

Intuitively, our theorem states that each convolution will cause the points in each cluster to "contract" towards their means up until a certain saturation point is reached. If after this contraction, many points are closer to their own centers than to any other centers, the softmax classifier will correctly classify them. Just as in Theorem 4.1, our error bound consists of one component depending on the variance, $\sigma^2$, that is large at the beginning and decreases exponentially with each convolution. The

classification accuracy at the point of saturation ($k \approx \log n$) depends on the squared product between graph's signal-to-noise ratio, $\delta/|\lambda|$, and the "separation ratio" of the datasets: $\|U\|_F / \Delta$. As in the two-class setting, our theorem captures both the homophilic and heterophilic settings. However, for large $L$, our error parameter, $\delta$, can be much larger if $q > p$. This observation of noisier graphs in the heterophilic setting, leading to less accurate performance, is consistent with observations from previous studies [Choi et al., 2023].

## 9   Experiments

In this section, we demonstrate our results empirically. For synthetic data, we show Theorems 4.1 and 4.2 for linear binary classification. For real data, we show that removing the principal component of the adjacency matrix exhibits positive effects on multi-class node classification problems as well.

### 9.1   Synthetic Data

For synthetic data from the CSBM, we demonstrate the benefits of removing the principal component of the adjacency matrix before performing convolutions for both variants of convolution described in Equation (1). We choose $n = 2000$ nodes with 20 features for each node, sampled from a Gaussian mixture. The intra-edge probability is fixed to $p = O(\log^3 n/n)$. We perform linear classification to demonstrate the results in Theorem 4.1 and Theorem 4.2, training a one-layer GCN network both with and without the corrected convolutions and perform an empirical comparison.

We provide plots for two different settings: (1) Fix $\gamma = |p - q|/(p + q) = 2/3$ and vary signal-to-noise ratio of the node features, $\|\mu - \nu\|/\sigma$, for different number of convolutions. We observe in Figure 1 that as the number of convolutions increases, the original GCN [Kipf and Welling, 2017] (in blue) starts performing poorly, while the corrected versions (in orange and green) retain the accuracy for lower signal-to-noise ratio; (2) Fix $\|\mu - \nu\|/\sigma = 1$ and vary the graph relative signal strength, $\gamma$, for different number of convolutions. We observe the same trends in this setting, as depicted in Figure 2. The vertical lines represent the threshold for exact classification from Theorem 4.2.

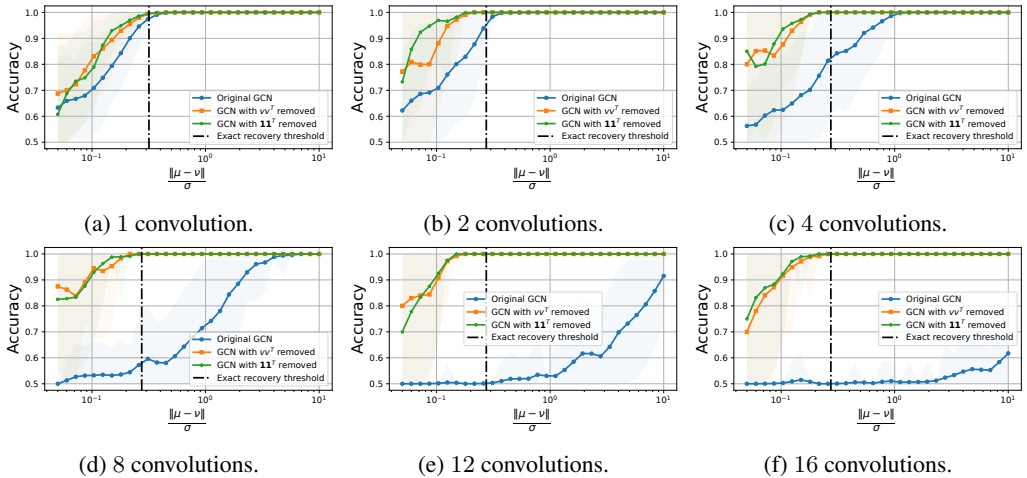

Figure 1: Accuracy plot (average over 50 trials) against the signal-to-noise ratio of the features (ratio of the distance between the means to the standard deviation) for increasing number of convolutions. Here, $v = D^{1/2}\mathbb{1}$ and the "GCN with $vv^\top$ removed" refers to convolution with the corrected, normalized adjacency matrix. "GCN with $\mathbb{1}\mathbb{1}^\top$ removed" is the corrected, unnormalized matrix.

### 9.2   Real Data

Similar to synthetic data, we compare the results for corrected graph convolution to the original GCN on the following real graph benchmarks datasets: *CORA*, *CiteSeer*, and *Pubmed* citation networks [Sen et al., 2008] in the multi-class setting. In Figure 3, we see that overall the accuracy of every learning method decreases as the number of convolutions increases but the corrected convolutions converge to an accuracy much higher than that of the uncorrected convolution. This is attributed to

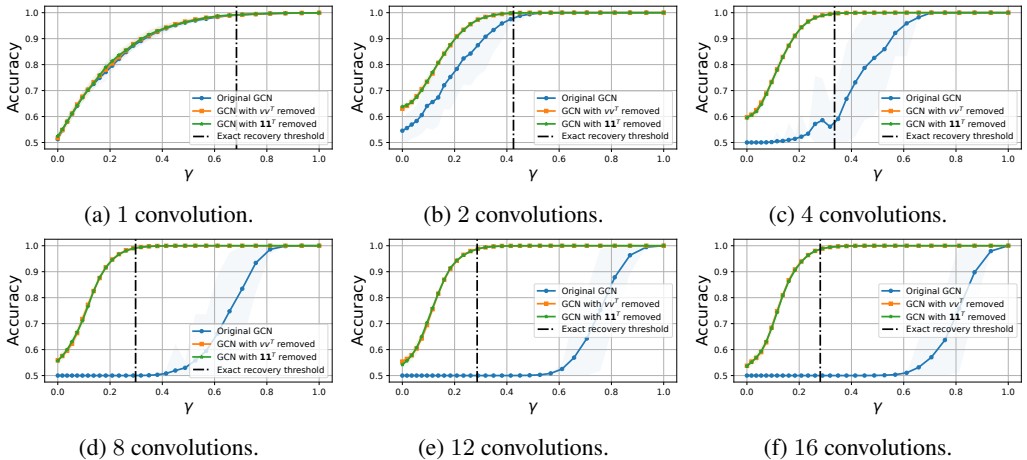

(a) 1 convolution.     (b) 2 convolutions.     (c) 4 convolutions.

(d) 8 convolutions.     (e) 12 convolutions.     (f) 16 convolutions.

Figure 2: Accuracy plot (average over 50 trials) against graph relative signal strength ($\gamma = |p - q|/(p + q)$) for various values of the number of convolutions.

the fact that for multi-class classification on general graphs, the important information about class memberships is typically captured by the top $C$ eigenvectors (except the first one) where $C$ is greater than the number of classes [Lee et al., 2014]. In general, these eigenvectors could have different eigenvalues. Since the limiting behavior of many rounds of convolutions is akin to projecting the features onto the eigenvector(s) corresponding to the second eigenvalue, we only expect this to capture partial information about the multi-class structure. By contrast, we show, in Appendix E.1, that for synthetic data with balanced classes, the classification accuracy only increases with more convolutions if they are corrected to remove the top eigenvector.

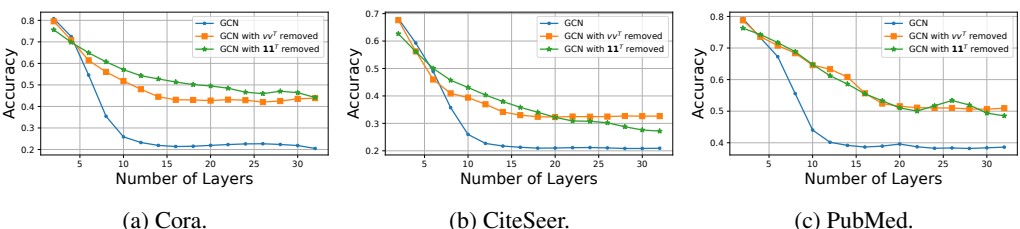

(a) Cora.       (b) CiteSeer.       (c) PubMed.

Figure 3: Accuracy plots (average over 50 trials) against the number of layers for real datasets.

## 10 Conclusion and Future Work

In this study, we utilized spectral methods to obtain partial and exact classification results for the linear classifier with corrected convolution matrices in the 2-block CSBM. Our spectral approach highlights, theoretically, how removing the top eigenvector can mitigate oversmoothing and improve classification accuracy. We prove that the removal of the top eigenvector results in reducing feature variance and correcting the asymptotic behavior of many rounds of convolution towards the second, rather than the top eigenvector of the adjacency matrix. Finally, we showed that our analysis can be generalized to the multi-class setting. We hope our analysis can lead to further developments in theoretical and practical studies of GNNs. A natural extension of this work would be to generalize our analysis to broader classes of multi-class models. For example, if the size of classes are unbalanced, the second eigenspace may not capture all the information about the clusters. In addition, the distribution of features may not follow a standard Gaussian mixture model, but more complicated distributions, possibly with multiple centers [Baranwal et al., 2023a]. Another natural setting to consider is when clusters in the feature distribution do not exactly match clusters in the graph. Extending our analysis to these settings will likely require more sophisticated network architectures and activation functions.

## Acknowledgements

K. Fountoulakis would like to acknowledge the support of the Natural Sciences and Engineering Research Council of Canada (NSERC). Cette recherche a été financée par le Conseil de recherches en sciences naturelles et en génie du Canada (CRSNG), [RGPIN-2019-04067, DGECR-2019-00147].

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

# A   Linear Algebra and Probability Background

Before beginning our proofs, we will establish the following theorems from the literature and basic facts that we will use throughout the proofs.

## A.1   Linear Algebra

For matrix inequalities, we use the following inequalities about matrix spectral norms.

**Theorem A.1.** *([Vershynin, 2018] theorem 4.5.3.) Let $A$ be a symmetric matrix with eigenvalues $\lambda_1 \geq \lambda_2 \geq ...\lambda_n$ and $B$ be a symmetric matrix $\mu_1 \geq \mu_2 \geq ....\mu_n$. Suppose $\|A - B\| \leq \delta$. Then, $\max_i |\lambda_i - \mu_i| \leq \delta$*

We also use the following basic result to relate the matrix spectral norm to its maximum entry

**Lemma A.2.** *Let $M$ be an $n \times n$ symmetric matrix such that each row of $M$ has at most $m$ non-zero entries and each entry of $M$ has an absolute value at most $\varepsilon$. Then $\|M\| \leq \varepsilon m$*

*Proof.* We will use the fact that for any scalars $a, b$, we have $2ab \leq a^2 + b^2$ since $a^2 + b^2 - 2ab = (a - b)^2 \geq 0$. We will also use $\text{supp}(M)$ to denote the set of non-zero entries in $M$. Let $x$ be a unit vector. Then we have

$$
\begin{aligned}
|x^\top M x| = |\sum_i \sum_{j=1}^n x(i)x(j)M_{i,j}| \\
\leq \varepsilon \sum_{i,j \in \text{supp}(M)} x(i)x(j) \\
\leq \varepsilon \sum_{i,j \in \text{supp}(M)} \frac{1}{2}x(i)^2 + \frac{1}{2}x(j)^2 \\
\leq \varepsilon m \sum_i x(i)^2 \\
= \varepsilon m
\end{aligned}
$$

Where the last inequality follows from the fact for every $i$, there are at most $m$ entries $j$ such that $M_{i,j} \neq 0$. $\square$

## A.2   Concentration Inequalities

Throughout our analysis, we will mainly use the following two concentration inequalities. The first is a bound on the the expected deviation of the sum of Bernoulli random variables

**Theorem A.3.** *Let $X_1, ...X_n$ be Bernoulli random variables with mean at most $p$ and $S_n = \sum_{i=1}^n X_i$. Then*

$$
\mathbf{Pr}\left[|S_n - \mathbb{E}[S_n]| > t\right] < \exp(-\Omega(\frac{t^2}{np}))
$$

The second concentration inequality we will upper-bounds the spectral norm of random matrices whose entries have zero mean and bounded variance.

**Theorem A.4.** *([Bandeira and van Handel, 2016] Remark 3.13) Let $R$ be a random matrix whose entries $R_{i,j}$ are independent, zero mean, with variance $\sigma^2$. Then with probability $1 - n^{-\Omega(1)}$, we have*

$$
\|R\| \leq O(\sigma\sqrt{n} + \log n)
$$

*In particular, if $\sigma^2 = \Omega(\frac{\log^2 n}{n})$ then $\|R\| \leq O(\sigma\sqrt{n})$*

We will also use the standard tail bound on the norm of a Gaussian vector

**Lemma A.5.** *Let $g \sim \mathcal{N}(0, \sigma^2 I_n)$ be a random Gaussian vector. Then $\mathbf{Pr}\left[\|g\| > t\right] \leq 2\exp(-\frac{t^2}{2n\sigma^2})$*

## A.3 Other Facts

We will also be using the following basic fact about approximating the exponential function

**Lemma A.6.** *For all $x$, $1 + x \leq e^x$. If $x \leq 1$, then $e^x \leq 1 + 2x$.*

# B  Proofs in Section 5

## B.1  Proof of Lemma 5.1

In the following section, we give the missing proofs of the basic properties of the CSBM. The first is the reduction from the $m-$dimensional model to the centered $1-$dimensional model, which we will restate here:

**Lemma B.1.** *Given an $m$-dimensional 2-block $CSBM$, there exist $w \in \mathbb{R}^m$ and $b \in \mathbb{R}^n$ such that $Xw + b = s + g$ where for each vertex $i$, $g_i$ is i.i.d. $\mathcal{N}(0, \sigma'^2)$ with $\sigma' = \frac{4\sigma}{\sqrt{n}\|\mu - \nu\|}$.*

*Proof.* Let $X_{i,:}$ be the $i^{th}$ row of $X$. For each $i \in S$, we have $X_{i,:} = \mu + g_i$ and for each $j \in T$ we have $X_{j,:} = \nu + g_j$, where $g_i \sim \mathcal{N}(0, \sigma^2 I_m)$ are i.i.d. random Gaussian noise vectors. Now we pick $w = (\mu - \nu)$, $b = -\frac{1}{2}\langle \mu + \nu, \mu - \nu \rangle \vec{\mathbb{1}}$. Now $x' = Xw + b$. For $i \in S$, we let

$$s'(i) := \langle \mu - \frac{1}{2}(\mu + \nu), \mu - \nu \rangle = \frac{1}{2}\|\mu - \nu\|^2$$

and for $i \in T$, we let

$$s'(i) := \langle \nu - \frac{1}{2}(\mu + \nu), \mu - \nu \rangle = -\frac{1}{2}\|\mu - \nu\|^2$$

The $s'$ part is our signal. On the other hand, the noise at each entry is given by $x'(i) - s'(i) = \langle g_i, \mu - \nu \rangle$ which is Gaussian with standard deviation $\sigma \|\mu - \nu\|$. Now, if we let $x = \frac{4}{\sqrt{n}\|\mu-\nu\|^2}x'$, then we have $x = s + g$, where $s(i) \in \pm\frac{1}{\sqrt{n}}$ and each entry of $g$ is i.i.d. Gaussian with standard deviation $\sigma' := \frac{4\sigma}{\sqrt{n}\|\nu-\mu\|}$. $\qquad\square$

**Proof of Proposition 5.2 and Characterizing the Convolution Matrix**

In this section, we will provide some crucial properties of the graph convolution matrices in (1) and then use them to prove Proposition 5.2. The adjacency matrix $A$ is a random matrix with expectation

$$\mathbb{E}[A] = \begin{bmatrix} pJ_{n/2} & qJ_{n/2} \\ qJ_{n/2} & pJ_{n/2} \end{bmatrix}$$

Where $J_m$ denotes the $m \times m$ all-ones matrix. It can be seen that $\mathbb{E}[A]$ is a rank-2 matrix, with eigenvectors $\frac{1}{\sqrt{n}}\mathbb{1}$ and $s$ because $\mathbb{E}[A]\mathbb{1} = \frac{1}{2}(p + q)n\mathbb{1}$ and $\mathbb{E}[A]s = \frac{1}{2}(p - q)ns$, which means

$$\mathbb{E}[A] = \frac{1}{2}(p + q)\mathbb{1}\mathbb{1}^\top + \frac{1}{2}(p - q)nss^\top$$

Thus, in the centered one-dimensional CSBM, our true signal, $s$, is encoded in two ways: as the second eigenvector of the matrix $\mathbb{E}[A]$ and as $\mathbb{E}[x]$. We can express our unnormalized corrected convolution matrix in terms of the signal, $ss^\top$, as follows:

$$\tilde{A} = \frac{1}{d}A - \frac{1}{n}\mathbb{1}\mathbb{1}^\top = \frac{(p - q)n}{2d}ss^\top + \frac{\frac{1}{2}(p + q)n - d}{nd}\mathbb{1}\mathbb{1}^\top + \frac{1}{d}(A - \mathbb{E}[A])$$

Recall from main text that we defined $\eta := \frac{(p-q)n}{2d}$ be the signal strength, $d' := \frac{(p+q)n/2-d}{nd}$ to be the degree deviation, and $R = A - \mathbb{E}[A]$ to be the error matrix. Thus, we have

$$\tilde{A} = \eta ss^\top + \frac{1}{d}R + d'\mathbb{1}\mathbb{1}^\top$$

Note that $R$ has independent zero mean entries with bounded variance, and such random matrices have well-studied concentration properties. The main idea of our analysis is to show that $d'$ and

$R$ are small compared to our matrix signal strength $\eta$ with high probability so that $\tilde{A}$ behaves like $\eta ss^\top \approx \gamma ss^\top$. To analyze convolution with the normalized $\hat{A}$, we use degree-concentration to show that $\hat{A}$ is close to $\tilde{A}$, and so also behaves like $\gamma ss^\top$.

Now we prove the main proposition in our section, Proposition 5.2, which we restate as follows:

**Proposition B.2.** *Assume that $p = \Omega(\frac{\log n}{n})$, and let $\gamma = \frac{p-q}{p+q}$. With probability $1 - n^{-\Omega(1)}$, we have the following concentration results*

1. $|d'| = \frac{|d - \frac{1}{2}(p+q)n|}{nd} \le O(1/n^{1.5})$, *which implies that* $\eta \in \frac{p-q}{p+q}(1 \pm o(1))$.

2. $\|R\| \le O(\sqrt{np})$

3. $\left\| \tilde{A} - \gamma ss^\top \right\| \le O(\frac{1}{\sqrt{np}})$ *and* $\left\| \hat{A} - \gamma ss^\top \right\| \le O(\sqrt{\frac{\log n}{np}})$

*Proof.* To bound the deviation of the average degree is equivalent to bounding the total number of edges in the graph. Note that the expected number of edges in the graph is $\frac{1}{4}(p+q)n^2$. Thus, we apply Theorem A.3 with $t = \Theta((p+q)n^{1.5})$ to obtain

$$\mathbf{Pr}\left[ ||E| - \frac{1}{4}(p+q)n^2| > t \right] \le \exp(-\Omega(pn)) = n^{-\Omega(1)}$$

Since $d = 2|E|/n$, we have that with high probability, $|d - \frac{1}{2}(p+q)n| \le O((p+q)\sqrt{n})$ and $|d'| = \frac{|d - \frac{1}{2}(p+q)n|}{nd} \le O(\frac{(p+q)\sqrt{n}}{(p+q)n^2(1-o(1))}) \le O(1/n^{1.5})$. This gives us bound number 1. To get bound number 2, we apply Theorem A.4 on the error matrix $R$, noting that each entry of $R$ has variance at most $p$. To get bound number 3, let $R' = \frac{1}{d}R + d'\mathbb{1}\mathbb{1}^\top = \tilde{A} - \eta ss^\top$. Then with high probability, we have

$$\|R'\| \le \frac{1}{d}\|R\| + d'\|J\|$$

$$\le \frac{1}{\frac{1}{2}(p+q)n(1-o(1))}\|R\| + d'\left\|\mathbb{1}\mathbb{1}^\top\right\|$$

$$\le O(\frac{1}{\sqrt{pn}} + \frac{1}{\sqrt{n}}) \quad \text{since } \left\|\mathbb{1}\mathbb{1}^\top\right\| = n$$

$$= O(\frac{1}{\sqrt{pn}})$$

Finally, we analyze the corrected normalized adjacency matrix $\hat{A}$. In this case, we want to show degree concentration for every node. Let $d_v$ be the degree of node $v$, and $\bar{d} = \frac{1}{2}(p+q)n$ be the expected degree. By applying Theorem A.3 and using our assumption $p > q$, we have

$$\mathbf{Pr}\left[ |d_v - \bar{d}| > \sqrt{Cnp\log n} \right] \le \exp(-\Omega(\log n))$$

Thus, with high probability, we have $|d_v - d| \le O(\sqrt{np\log n})$ for all $v \in V$. Now, to analyze the normalized adjacency matrix, we have

$$\left\| \hat{A} - \gamma ss^\top \right\| = \left\| \hat{A} - \gamma ss^\top + \tilde{A} - \tilde{A} \right\|$$

$$\le \left\| \hat{A} - \tilde{A} \right\| + \left\| \tilde{A} - \gamma ss^\top \right\|$$

$$\le \left\| D^{-1/2}AD^{-1/2} - \frac{1}{d}A \right\| + \left\| \frac{1}{\sum_v d_v}D^{1/2}\mathbb{1}\mathbb{1}^\top D^{1/2} - \frac{1}{n}\mathbb{1}\mathbb{1}^\top \right\| + O(\frac{1}{\sqrt{np}})$$

To bound $\left\| D^{-1/2}AD^{-1/2} - \frac{1}{d}A \right\|$, let $\varepsilon = \sqrt{\frac{C\log n}{np}}$ for large enough $C$ so that for all $v$, $d_v \in \bar{d}(1 \pm \varepsilon)$. We note that the adjacency matrix has at most $\bar{d}(1+\varepsilon)$ non-zero entries for each row and for each $u, v$, we have

$$|\frac{1}{\sqrt{d_u d_v}} - \frac{1}{d}| \le \frac{1}{\bar{d}(1-\varepsilon)} - \frac{1}{\bar{d}(1+\varepsilon)} \le O(\frac{\varepsilon}{d})$$

Thus, by applying Lemma A.2, we have $\left\| D^{-1/2}AD^{-1/2} - \frac{1}{d}A \right\| \leq O(\varepsilon)$. Similarly, for all $u, v$, we have

$$|\frac{\sqrt{d_u d_v}}{\sum_i d_i} - \frac{1}{n}| \leq \frac{\bar{d}(1+\varepsilon)}{\bar{d}n(1-\varepsilon)} - \frac{1}{n} \leq O(\frac{\varepsilon}{n})$$

Thus, Lemma A.2 implies that $\left\| \frac{1}{\sum_v d_v}D^{1/2}\mathbb{1}\mathbb{1}^\top D^{1/2} - \frac{1}{n}\mathbb{1}\mathbb{1}^\top \right\| \leq O(\varepsilon)$. Finally, this implies that

$$\left\| \hat{A} - \gamma ss^\top \right\| \leq O(\sqrt{\frac{\log n}{np}})$$

$\square$

Note that the case $p > q$ corresponds to a homophilous graph, and the case $p < q$ corresponds to a heterophilous graph (see Luan et al. [2021], Ma et al. [2022] for more), however, for binary classification, it has been shown [Baranwal et al., 2023a] that once can assume $p > q$ without loss of generality, and make corresponding adjustments in the classifier. Indeed in our case, if $p < q$ then we can take $-\tilde{A}$ as our convolution matrix so that its maximum modulus eigenvalue is positive. Moreover, since our signal, $s$, is centered, taking a convolution with $\tilde{A}$ is equivalent to taking a convolution with $-\tilde{A}$. Thus, we can assume without loss of generality that $\gamma$ is always non-negative, i.e. $p > q$ for the rest of our analysis.

## C  Proofs in Section 6

In this section, we will formally prove our first main theorem on partial classification, Theorem 4.1. To start, we will show the correlation between the top eigenvector of the convolution matrix $M$ and $s$. Note that this result is well-known in the literature (for example see Vershynin [2018] chapter 4.5). We will give the proof here for completeness.

**Lemma C.1.** *Suppose $M$ satisfies $\left\| M - \gamma ss^\top \right\| \leq \delta$ where $\gamma > C\delta$ for large enough constant $C$. Let $\hat{s}$ be the top eigenvector of $M$. Then $\langle s, \hat{s} \rangle^2 \geq 1 - 4\frac{\delta^2}{\gamma^2}$, and $\|s - \hat{s}\|^2 = O(\delta/\gamma)$.*

*Proof.* By assumption, we can express $M$ as $\gamma ss^\top + R'$ where $\|R'\| \leq \delta$. Now let $\lambda_k$ be the $k^{th}$ largest eigenvalue of $M$. By Theorem A.1, we have $|\lambda_1 - \gamma| \leq \delta$ and $\forall i > 1, |\lambda_i| \leq \delta$. Now consider the matrix $I - ss^\top$, which is the projection matrix onto the orthogonal complement subspace of $s$. Since $M = \gamma ss^\top + R'$, we have

$$\gamma(I - ss^\top) = \gamma I - M + R'$$

and the matrix $\gamma I - M$ has the same eigenvectors as $M$. In particular, $\hat{s}$ is an eigenvector with eigenvalues $\gamma - \lambda_1$. Thus, we have

$$\begin{aligned}
\gamma\sqrt{1 - \langle s, \hat{s}\rangle^2} = \gamma\sqrt{\hat{s}^\top(I - ss^\top)\hat{s}} &= \left\| \gamma(I - ss^\top)\hat{s} \right\| \\
&= \|(\gamma I - M)\hat{s} + R'\hat{s}\| \\
&\leq \|(\gamma I - M)\hat{s}\| + \|R'\hat{s}\| \\
&\leq |\gamma - \lambda_1| + \delta \\
&\leq 2\delta
\end{aligned}$$

Thus, we have $\langle s, \hat{s}\rangle^2 \geq 1 - 4\delta^2/\gamma^2$. Since by our assumption, $\delta/\gamma$ is small, $\hat{s}$ is very close to either $s$ or $-s$. For our analysis, it doesn't matter which of these is the case as both $s$ and $-s$ equally separate the points in our two classes. Thus, we can assume WLOG that $\langle s, \hat{s}\rangle > 0$. As a consequence, we have

$$\|s - \hat{s}\|^2 = 2 - 2\langle s, \hat{s}\rangle = 2 - 2\sqrt{1 - 2\delta/\gamma} = O(\delta/\gamma)$$

where the second inequality follows from applying the first order approximation $\sqrt{1 - x} = 1 - \frac{x}{2} - O(x^2)$ using the fact that $\delta/\gamma$ is small. Note that if $\langle s, \hat{s}\rangle$ was negative, we can do the same analysis in the following sections with $-s$ instead of $s$. $\square$

Now we will prove the main result about partial classification in the 1-dimensional model, Proposition 6.1, which we restate as follows:

**Proposition C.2.** *Given a centered 1-dimensional 2 block CSBM with parameters $n, p > q, \sigma$ and $\gamma(p, q) = \frac{p-q}{p+q}$, suppose our convolution matrix $M$ satisfies $\left\| M - \gamma s s^\top \right\| \leq \delta$ and $\gamma \geq C\delta$ for a large enough constant $C$. Let $x_k = M^k x$, be the result of applying $k$ rounds of graph convolution to our input feature $x$. Then with probability at least $1 - \frac{1}{2} \exp(-\frac{1}{4\sigma^2})$, there exists a scalar $C_k$ and an absolute constant $C'$ such that*

$$\|C_k x_k - s\|^2 \leq O\left(\frac{\delta^2}{\gamma^2} + \left(\frac{C'\delta}{\gamma}\right)^{2k} n\sigma^2 \log n\right)$$

*Proof.* Let $\lambda_1 \geq \ldots \lambda_n$ be the eigenvalues of $M$ and $w_1 =: \hat{s}, w_2 \ldots w_n$ be their corresponding eigenvectors. By Lemma C.1, the eigenvalues satisfy $\lambda_1 \in \gamma \pm \delta$ and $|\lambda_i| \leq \delta$ for $i > 1$. The convolution vector after $k$ rounds can be expressed as $x_k = M^k g + M^k s$, and we will analyze each term individually.

$$M^k g = \lambda_1^k \langle g, w_1 \rangle \hat{s} + \sum_{i>1} \lambda_i^k \langle g, w_i \rangle w_i$$
$$= \lambda_1^k \langle g, w_1 \rangle s + \lambda_1^k \langle g, w_1 \rangle (\hat{s} - s) + \sum_{i>1} \lambda_i^k \langle g, w_i \rangle w_i$$

and similarly,

$$M^k s = \lambda_1^k \langle s, w_1 \rangle \hat{s} + \sum_{i>1} \lambda_i^k \langle s, w_i \rangle w_i$$
$$= \lambda_1^k \langle s, w_1 \rangle s + \lambda_1^k \langle s, w_1 \rangle (\hat{s} - s) + \sum_{i>1} \lambda_i^k \langle s, w_i \rangle w_i$$

Now we will let $E_g$ and $E_s$ be the error vectors, i.e. vectors that are not in the span of $s$, from each of the terms respectively. That is

$$E_g = \lambda_1^k \langle g, w_1 \rangle (\hat{s} - s) + \sum_{i>1} \lambda_i^k \langle g, w_i \rangle w_i$$
$$E_s = \lambda_1^k \langle s, w_1 \rangle (\hat{s} - s) + \sum_{i>1} \lambda_i^k \langle s, w_i \rangle w_i$$

Thus, we have

$$x_k = \lambda_1^k \langle s + g, \hat{s} \rangle s + E_g + E_s$$

Thus, we will let $C_k = (\lambda_1^k \langle s + g, \hat{s} \rangle)^{-1}$ and bound the squared norm of the error terms $C_k (E_g + E_s)$. We will also use the fact that for any vectors $y, z$, we have $\|y + z\|^2 \leq 2 \|y\|^2 + 2 \|z\|^2$. Using this, we have

$$\|E_g\|^2 \leq 2\lambda_1^{2k} |\langle \hat{s}, g \rangle|^2 \|s - \hat{s}\|^2 + 2 \left\| \sum_{i>1} \lambda_i^k \langle g, w_i \rangle w_i \right\|^2$$
$$\leq O(\lambda_1^{2k} \delta^2 / \gamma^2) |\langle \hat{s}, g \rangle|^2 + 2\delta^{2k} \left\| \sum_{i>1} \langle g, w_i \rangle w_i \right\|^2$$
$$\leq O(\lambda_1^{2k} \delta^2 / \gamma^2) |\langle \hat{s}, g \rangle|^2 + 2\delta^{2k} \|g\|^2$$

where in the second inequality, we used the fact that Lemma C.1 implies $\|s - \hat{s}\|^2 \leq O(\delta^2/\gamma^2)$. By Lemma A.5, we have $\|g\|^2 = O(\sigma^2 n \log n)$ with probability $1 - n^{-\Omega(1)}$. Also, $\langle \hat{s}, g \rangle$ has distribution $\mathcal{N}(0, 1)$ because $\hat{s}$ is a unit vector. Thus with probability $\exp(-\frac{1}{4\sigma^2})$, $|\langle \hat{s}, g \rangle| \leq 1/2$. If both these events happen, then we have $\|E_g\|^2 \leq O(\lambda_1^{2k} \delta^2/\gamma^2 + \delta^{2k} \sigma^2 n \log n)$. Now, to bound the other error term, we have

$$\|E_s\|^2 \leq 2\lambda_1^{2k} \|s - \hat{s}\|^2 \langle \hat{s}, s \rangle^2 + 2 \left\| \sum_{i>1} \lambda_2^k \langle w_i, s \rangle w_i \right\|^2$$
$$\leq 2\lambda_1^{2k} \|s - \hat{s}\|^2 \langle \hat{s}, s \rangle^2 + 2\delta^{2k} \|s\|^2$$
$$\leq O(\lambda_1^{2k} \delta^2/\gamma^2),$$

where the last inequality follows from the fact that $\|s\| = 1$ and $|\lambda_1 - \gamma| \leq \delta$, which is a constant factor smaller than $\lambda_1$, which means $\delta^{2k} = O(\lambda_1^{2k}\delta^2/\gamma^2)$. Finally, we want to lower bound $\langle \hat{s}, s \rangle + \langle s, g \rangle$ by a constant so $C_k = O(\lambda_1^k)$. By Lemma C.1, we have $\langle \hat{s}, s \rangle > 1 - O(\delta/\gamma)$. We can assume that our constant $C$ is large enough so that $\gamma/\delta < 1/4$, which means that $\langle \hat{s}, s \rangle \geq 3/4$. Then $\langle \hat{s}, s \rangle + \langle s, g \rangle \geq 1/4$ as long as $\langle s, g \rangle \geq 0$ or $|\langle s, g \rangle| \leq 1/2$. This happens with probability at least $\frac{1}{2} + \frac{1}{2}(1 - \exp(-\frac{1}{4\sigma^2})) = 1 - \frac{1}{2}\exp(-\frac{1}{4\sigma^2})$, and if this does occur, we have

$$\|C_k x_k - s\|^2 \leq 2C_k^2 \|E_g\|^2 + 2C_k^2 \|E_s\|^2$$
$$\leq O(\lambda_1^{2k}(\lambda_1^{-2k}\delta^2/\gamma^2 + \delta^{2k}n\sigma^2 \log n))$$
$$O(\delta^2/\gamma^2 + (\delta/\lambda_1)^{2k}n\sigma^2 \log n)$$

Moreover, since $\lambda_1 \geq \gamma - \delta$, we have $\delta/\lambda_1 \leq \delta/(\gamma - \delta) = O(\delta/\gamma)$ since $\gamma$ is a least a constant factor larger than $\delta$, and this gives us our final bound. $\qquad \square$

With our main proposition established, we can prove our main theorem as follows:

*Proof.* (Theorem 4.1) First, we apply Lemma 5.1 to reduce to the one dimensional centered CSBM with parameters $p, q$ and $\sigma' = O(\frac{\sigma}{\sqrt{n}\|\mu - \nu\|})$. Now let $x = Xw + b$ be our transformed one-dimensional feature vector and let $x_k = M^k x$ where $M$ is either $\tilde{A}$ or $\hat{A}$. To bound the mean-squared error of our convolved data, we will apply Proposition 6.1. By Proposition 5.2, we can take $\delta = O(\frac{1}{\sqrt{np}})$ if $M = \tilde{A}$ and $\delta = O(\sqrt{\frac{\log n}{np}})$ if $M = \hat{A}$. Then, as long as $\gamma \geq C\delta$ for large enough constant $C$, we have a scalar $C_k$ and constant $C'$ such that

$$\|C_k x_k - s\|^2 \leq O\left(\frac{\delta^2}{\gamma^2} + \left(\frac{C'\delta}{\gamma}\right)^{2k}\frac{\sigma^2 \log n}{\|\mu - \nu\|^2}\right)$$

Now, we take $0$ to be the threshold, meaning we put vertex $i$ in the first class if $x_k(i) < 0$ and put it in the second class otherwise. Note that this partitioning scheme is indifferent to the scaling of the vector $x_k$, so we can equivalently apply it to $C_k x_k$ (note this does not necessitate computing $C_k$ directly). Since each $i \in S$ has $s(i) = 1/\sqrt{n}$ and each $j \in T$ has $s(j) = -1/\sqrt{n}$, a vertex can only be misclassified if it contributes at least $1/n$ to the total squared distance $\|C_k x_k - s\|^2$. Thus, the total number of misclassified vertices is at most $\|C_k x_k - s\|^2 n$, which gives us the main theorem after substituting the appropriate value of $\delta$. $\qquad \square$

## D   Proofs in Section 7

In this section, we will formally prove Proposition 7.1. Before we begin, we will, as a warm-up, analyze the behavior of 1 convolution for the centered 1-dimensional CSBM. The following proposition is essentially equivalent to theorem 1.2 in Baranwal et al. [2021]. The proof here is not used later on, but gives some intuition about how to analyze exact classification using a matrix-focused framework.

**Proposition D.1.** *Suppose we are given the 1-dimensional centered CSBM with parameters $n, p, q, \sigma$ such that $p \geq \Omega(\frac{\log^2 n}{n})$, $\gamma \geq \Omega(\sqrt{\frac{\log n}{np}})$, and $\sigma \leq O(\frac{1}{\sqrt{\log n}})$. Then with probability $1 - n^{-\Omega(1)}$, we have*

$$\left\|s - \frac{1}{\eta}x_1\right\|_\infty \leq O\left(\sigma\left(\frac{1}{\gamma}\sqrt{\frac{\log n}{np}} + \sqrt{\frac{\log n}{n}}\right)\right) + \frac{1}{2\sqrt{n}}$$

*Proof.* By the definition of our convolution matrix, we have

$$\frac{1}{\eta}x_1 = \frac{1}{\eta}(\eta ss^\top + R')(s + g) = s + \frac{1}{\eta}R's + (ss^\top + \frac{1}{\eta}R')g$$

Thus, we can bound the infinity norm error as:

$$\left\|s - \frac{1}{\eta}x_1\right\|_\infty = \left\|\frac{1}{\eta}R's + (ss^\top + \frac{1}{\eta}R')g\right\|_\infty \leq \left\|(ss^\top + \frac{1}{\eta}R')g\right\|_\infty + \left\|\frac{1}{\eta}R's\right\|_\infty$$

Since the infinity norm is the maximum absolute value of all entries, we need to bound terms $|e_u^\top R's|/\eta$ and $|e_u^\top(ss^\top + R'/\eta)g|$ for all $u \in V$. We will start by bounding the Gaussian part of the error. Since $e_u^\top(ss^\top + R'/\eta)g \sim \mathcal{N}(0, \sigma^2 \|R'e_u/\eta + \langle s, e_u\rangle s\|^2)$, we have that with high probability $\max_{i \in n} |e_u^\top(ss^\top + R'/\eta)g| \leq O(\sigma\sqrt{\log n} \cdot \|R'e_u/\eta + \langle s, e_u\rangle s\|)$. Now note that $|\langle s, e_u\rangle| = 1/\sqrt{n}$ and by Theorem A.4, $\|R'e_u\| \leq \|R'\| \leq O(\sigma\sqrt{\frac{1}{np}})$. Finally, by Proposition 5.2, we have $\eta = \gamma(1 \pm o(1))$. Thus, we have

$$|e_u^\top(ss^\top + R'/\eta)g| \leq O(\sigma\Big(\frac{1}{\gamma}\sqrt{\frac{\log n}{np}} + \sqrt{\frac{\log n}{n}}\Big))$$

Now, to bound the error term $|e_u^\top R's|$, we have $e_u^\top R's = e_u^\top(\frac{1}{d}R + d'\mathbb{1}\mathbb{1}^\top)s = \frac{1}{d}e_u^\top Rs$ since $s$ is orthogonal to the all-ones vector. Now, WLOG, suppose $u \in S$. Notice that $e_u^\top Rs = \frac{1}{\sqrt{n}}(\sum_{i \in S} R_{u,i} - \sum_{i \in T} R_{u,i})$, where both sums are sums over independent shifted Bernoulli random variables with variance at most $p$. Thus, by applying Theorem A.3, we have $|e_u^\top R's| \leq \frac{1}{\sqrt{n}} \cdot O(\sqrt{np\log n}) = O(\sqrt{p\log n})$. Now, using the fact that $d \geq \frac{1}{2}(p+q)n(1 - o(1))$, we have $|\frac{1}{d}e_u^\top Rs| \leq O(\frac{1}{\sqrt{n}} \cdot \sqrt{\frac{\log n}{np}})$. By our assumption, $\gamma \geq C\sqrt{\frac{\log n}{np}}$) for large enough constant $C$. Then, we can take $C$ to be large enough so that

$$|\frac{1}{\eta}e_u^\top R's| = O\Big(\frac{1}{\gamma(1 - o(1))}\sqrt{\frac{\log n}{np}}\Big) \leq \frac{1}{2\sqrt{n}}$$

$\square$

### D.1 Proof of Main Result

In this section, we prove Proposition 7.1, which we restate as follows:

**Proposition D.2.** *Suppose we are given a 1-dimensional centered 2-block CSBM with parameters* $n, p, q, \sigma$ *and* $k = O(\log n)$ *such that* $\gamma = \frac{p-q}{p+q} \geq \Omega(k\sqrt{\frac{\log n}{np}})$, $p \geq \frac{\log^3 n}{n}$, *and* $\sigma \leq O(\frac{1}{\sqrt{\log n}})$. *Then with probability at least* $1 - n^{-\Omega(1)}$:

$$\left\|\frac{1}{\eta^k}\tilde{A}^k x - s\right\|_\infty \leq \frac{1}{2\sqrt{n}} + O((\frac{C}{\gamma\sqrt{np}})^k \sigma\sqrt{\log n})$$

Before proving the main proposition, we will use degree-concentration to reduce analyzing the error matrix $R' = \tilde{A} - \eta ss^\top$ to analysing $\frac{1}{d}R$ instead. This is useful because our analysis crucially uses the fact that our error matrix $R$ has zero-mean Radamacher entries.

**Proposition D.3.** *Let* $R' = \tilde{A} - \eta ss^\top$ *and suppose* $k \leq O(\log n)$. *Then with high probability, we have that* $\|R'^k - \frac{1}{d^k}R^k\| \leq O(\frac{C^k k\sqrt{\log n}}{(\sqrt{pn})^k \cdot \sqrt{n}})$ *for a constant* $C$.

*Proof.* Before beginning the proof, we first need to give a tighter degree-concentration bound to show that with high probability, $|d'| \leq O(\frac{\sqrt{\log n}}{n^2\sqrt{p}})$. By applying Theorem A.3 with $t = \Theta(n\sqrt{p\log n})$.

$$\mathbf{Pr}\left[||E| - \frac{1}{4}(p+q)n^2| > t\right] \leq \exp(-\Omega\Big(\frac{n^2p\log n}{(p+q)n^2}\Big)) = n^{-\Omega(1)}$$

Since $d = 2|E|/n$ and with high probability at, $|E| \geq \frac{1}{4}(p+q)n^2 - t \geq \frac{1}{4}(p+q)n^2(1 - o(1))$, we have $d \geq \frac{1}{2}(p+q)n(1 - o(1)) = \Omega(np)$. Putting these together, we get the bound:

$$|d'| = \frac{|d - \frac{1}{2}(p+q)n|}{nd} = 2\frac{||E| - \frac{1}{4}(p+q)n^2|}{n^2d} \leq O(\frac{n\sqrt{p\log n}}{n^3p}) \leq O(\frac{\sqrt{\log n}}{n^2\sqrt{p}})$$

Now, recall from Equation (2) that $R' = \frac{1}{d}R + d'\mathbb{1}\mathbb{1}^\top$. By Proposition 5.2, we have that with high probability, $\frac{1}{d}\|R\| \leq \sqrt{\frac{C}{np}}$ for some constant $C$. We will let $\delta := \sqrt{\frac{C}{np}}$ be this upper bound. Now

consider the matrix $(\frac{1}{\delta}R')^k$. We will let $M_0 = \frac{1}{d\delta}R$ and $M_1 = \frac{d'}{\delta}\mathbb{1}\mathbb{1}^\top$. Then, we have $\|M_0\| \le 1$, and $\|M_1\| \le \frac{d'n}{\delta}$. Since $|d'| \le \frac{\sqrt{\log n}}{n^2\sqrt{p}}$ with high probability, we will assume this event occurs. Thus, we have $\|M_1\| \le O(\frac{n\sqrt{np\log n}}{n^2\sqrt{p}}) \le O(\sqrt{\frac{\log n}{n}})$. Finally, we have $(\frac{1}{\delta}R')^k = (M_0 + M_1)^k$. We are going to expand this out, and so let us introduce some notation. Let $\binom{[k]}{\ell} = \{i := (i_1, i_2, ...i_k) \in \{0,1\}^k : i_1 + i_2 + ...i_k = \ell\}$. That is, it's the set of length $k$ binary sequences with exactly $\ell$ terms equaled to 1. Now, we have

$$(M_0 + M_1)^k - M_0^k = \sum_{\ell=0}^{k} \sum_{i \in \binom{[k]}{\ell}} \prod_{j=1}^{k} M_{i_j} - M_0^k$$

$$= \sum_{\ell=1}^{k} \sum_{i \in \binom{[k]}{\ell}} \prod_{j=1}^{k} M_{i_j}$$

For any matrices $M, N$, their spectral norms satisfy $\|MN\| \le \|M\| \|N\|$. Thus, for each $i \in \binom{[k]}{\ell}$, we have $\left\|\prod_{j=1}^{k} M_{i_j}\right\| \le \|M_1\|^\ell$ since $\|M_0\| \le 1$. Thus, we have

$$\left\|(M_0 + M_1)^k - M_0^k\right\| \le \sum_{\ell=1}^{k} \binom{k}{\ell} \|M_1\|^\ell$$

$$= (1 + \|M_1\|)^k - 1$$

$$\le e^{k\|M_1\|} - 1$$

$$\le 2k\|M_1\|$$

where the second last inequality follows from Lemma A.6 because $k\|M_1\| \le k\sqrt{\frac{\log n}{n}} \le 1$ by our assumptions on $k$. Finally, we have

$$\left\|R'^k - \frac{1}{d^k}R^k\right\| = \delta^k \left\|(M_0 + M_1)^k - M_0\right\|$$

$$\le 2\delta^k k \|M_1\|$$

$$\le O\left(\frac{kC^k\sqrt{\log n}}{(\sqrt{pn})^k \cdot \sqrt{n}}\right)$$

For a constant $C$. $\qquad\square$

As a corollary, we can show the following:

**Corollary D.4.** *Let $x$ and $y$ be unit vectors. Then $|x^\top R'^k y| \le |x^\top(\frac{1}{d^k}R^k)y| + O(\frac{C^k k\sqrt{\log n}}{(\sqrt{pn})^k \cdot \sqrt{n}}))$ for some constant $C$.*

*Proof.* This follows from the basic properties of the spectral norm

$$|x^\top R'^k y| = |x^\top(\frac{1}{d^k}R^k + R'^k - \frac{1}{d^k}R^k)y|$$

$$\le |x^\top(\frac{1}{d^k}R^k)y| + \|x\| \|y\| \left\|R'^k - \frac{1}{d^k}R^k\right\|$$

$$\le |x^\top(\frac{1}{d^k}R^k)y| + O(\frac{C^k k\sqrt{\log n}}{(\sqrt{pn})^k \cdot \sqrt{n}})) \quad \text{by Proposition D.3}$$

$\qquad\square$

Corollary D.4 is key to proving our main technical lemma, the proof of which we will defer to the next section.

**Lemma D.5.** *Given $p \geq \frac{\log^3 n}{n}$, we have that with probability $1 - n^{-\Omega(1)}$, for all $u \in V$ and all $k \in \{1, 2, \ldots O(\log n)\}$*

$$|e_u^\top R'^k s| \leq \frac{1}{\sqrt{n}} \left( C \sqrt{\frac{\log n}{pn}} \right)^k$$

*for some constant $C$*

Now, we are ready to prove our main result, Proposition 7.1. The proof is similar to that of Proposition D.3 but requires the use of Lemma D.5 to obtain a sharper bound on the error terms than by simply applying the spectral norm bound as we did in the degree-concentration proof.

*Proof.* (Proposition 7.1) Just like in the $k = 1$ case, we express $\tilde{A}$ as $\eta s s^\top + R'$ and decompose our convolution vector into the following terms:

$$\frac{1}{\eta^k} \tilde{A}^k x = \frac{1}{\eta^k} \tilde{A}^k (s + g)$$

$$= s + \left( \frac{1}{\eta^k} \tilde{A}^k - s s^\top \right) s + \frac{1}{\eta^k} \tilde{A}^k g$$

In order to bound $\left\| s - \frac{1}{\eta^k} \tilde{A}^k x \right\|_\infty$, we will bound, for all $u \in V$, the error terms $|e_u^\top (\frac{1}{\eta^k} \tilde{A}^k - s s^\top) s|$ and $|e_u^\top \frac{1}{\eta^k} \tilde{A}^k g|$. Similar to in the proof of Proposition D.3, we start by expanding out $\frac{1}{\eta^k} \tilde{A}^k$. Let $Y_0 = s s^\top$ and $Y_1 = R'$. Then we have

$$\frac{1}{\eta^k} \tilde{A}^k = \frac{1}{\eta^k} (\eta Y_0 + Y_1)^k$$

$$= \frac{1}{\eta^k} \sum_{\ell=0}^k \eta^{k-\ell} \sum_{i \in \binom{[k]}{\ell}} \prod_{j=1}^k Y_{i_j}$$

$$= \sum_{\ell=0}^k \eta^{-\ell} \sum_{i \in \binom{[k]}{\ell}} \prod_{j=1}^k Y_{i_j}$$

Note that $(s s^\top)^2 = s s^\top$, which means that the first term in our summation (i.e. the $\ell = 0$ terms) is simply $s s^\top$. Now we start by bounding the error terms involving $s$.

$$|e_u^\top (\frac{1}{\eta^k} \tilde{A}^k - s s^\top) s| = |\sum_{\ell=1}^k \eta^{-\ell} \sum_{i \in \binom{[k]}{\ell}} e_u^\top \prod_{j=1}^k Y_{i_j} s|$$

$$\leq \sum_{\ell=1}^k \eta^{-\ell} \sum_{i \in \binom{[k]}{\ell}} |e_u^\top \prod_{j=1}^k Y_{i_j} s|$$

For each $i \in \binom{[k]}{\ell}$, we have

$$|e_u^\top \prod_{j=1}^k Y_{i_j} s| = |e_u^\top R'^{a_1} s \cdot s^\top R'^{a_2} s \cdot \ldots s^\top R'^{a_L} s|$$

where $a_1, \ldots a_L$ are non-negative integers satisfying $a_1 + \ldots a_L = \ell$. This is because the product $\prod_{j=1}^k Y_{i_j}$ has exactly $\ell$ terms of the form $R'$ and $s s^\top$ for the rest of the terms. By Proposition 5.2, the matrix $R'$ has spectral norm at most $\frac{C_1}{\sqrt{np}}$ for some constant $C_1$ with high probability. For each term of the form $|s^\top R'^{a_j} s|$ for $j > 1$, we bound it by $|s^\top R'^{a_j} s| \leq \|R'\|^{a_j} \leq \left( \frac{C_1}{\sqrt{np}} \right)^{a_j}$. For the first term $|e_u^\top R'^{a_1} s|$, we apply Lemma D.5 and give $|e_u^\top R'^{a_1} s| \leq \frac{1}{\sqrt{n}} \left( C_2 \sqrt{\frac{\log n}{np}} \right)^{a_1}$ for some constant $C_2$. Thus, we have:

$$|e_u^\top \prod_{j=1}^k Y_{i_j} s| \leq \frac{1}{\sqrt{n}} \left( \frac{\max(C_1, C_2)}{\sqrt{np}} \right)^{a_1 + \ldots a_L} \sqrt{\log n}^{a_1} \leq \frac{1}{\sqrt{n}} \left( C \sqrt{\frac{\log n}{np}} \right)^\ell$$

where $C$ is a large enough constant such that $C \geq \max(C_1, C_2)$. Now, we let $\rho := \frac{C}{\eta}\sqrt{\frac{\log n}{np}}$. Assuming that $\gamma \geq 9Ck\sqrt{\frac{\log n}{np}}$ and with high probability, $\eta \in \gamma(1 \pm o(1))$, we have $\rho \leq \frac{1}{8k}$ with high probability. Thus, we have

$$
\begin{aligned}
|e_u^\top(\frac{1}{\eta^k}\tilde{A}^k - ss^\top)s| &\leq \sum_{\ell=1}^{k} \eta^{-\ell} \sum_{i \in \binom{[k]}{\ell}} |e_u^\top \prod_{j=1}^{k} Y_{i_j} s| \\
&\leq \frac{1}{\sqrt{n}} \sum_{\ell=1}^{k} \binom{k}{\ell} \eta^{-\ell} \left(C\sqrt{\frac{\log n}{np}}\right)^\ell \\
&= \frac{1}{\sqrt{n}} \sum_{\ell=1}^{k} \binom{k}{\ell} \rho^\ell \\
&= \frac{1}{\sqrt{n}}((1+\rho)^k - 1) \\
&\leq \frac{1}{4\sqrt{n}} \quad \text{by Lemma A.6 and } \rho \leq \frac{1}{8k}
\end{aligned}
$$

Now, we bound the Gaussian part of the error:

$$
e_u^\top \tilde{A}g = \sum_{\ell=0}^{k-1} \eta^{-\ell} \sum_{i \in \binom{[k]}{\ell}} e_u^\top \prod_{j=1}^{k} Y_{i_j} g + \eta^{-k} e_u^\top R'^k g
$$

For each $i \in \binom{[k]}{\ell}$ where $\ell < k$, we have

$$
|e_u^\top \prod_{j=1}^{k} Y_{i_j} g| = |e_u^\top R'^{a_1} s \cdot s^\top R'^{a_2} s \cdot ... s^\top R'^{a_L} g|
$$

where $a_1 + a_2 + ... a_L = \ell$. This is because when $\ell < k$, there is at least one $Y_{i_j}$ term that is equalled to $ss^\top$, which means in the product, the leftmost term must be $e_u^\top R'^{a_1} s$ for some $a_1$ and the right most term must be $s^\top R'^{a_L} g$ for some $a_L$. Once again, we use the fact that $|s^\top R'^{a_j} s| \leq \|R'\|^{a_j} \leq (\frac{C_1}{\sqrt{np}})^{a_j}$ for all $j > 1$, and by Lemma D.5, we have $|e_u^\top R'^{a_1} s| \leq (C_2\sqrt{\frac{\log n}{np}})^{a_1}$ where $C_1, C_2$ are absolute constants. To bound the last term, we have that $s^\top R'^a g \sim \mathcal{N}(0, \sigma^2 \|R'^a s\|^2)$ for all $a > 0$. Thus, with high probability, we have that $|s^\top R'^{a_L} g| \leq O(\sigma \|R'^{a_L} s\| \sqrt{\log n}) \leq O(\sigma(\frac{C_1}{\sqrt{np}})^{a_L} \sqrt{\log n})$. Thus, we can apply the same bounds as we did for the error term involving $s$ but now with an extra $\sigma\sqrt{\log n}$ factor:

$$
\begin{aligned}
|e_u^\top \prod_{j=1}^{k} Y_{i_j} g| = |e_u^\top R'^{a_1} s \cdot s^\top R'^{a_2} s \cdot ... s^\top R'^{a_L} g| \\
\leq O(\frac{\sigma\sqrt{\log n}}{\sqrt{n}} \cdot \sum_{\ell=0}^{k-1} \binom{k}{\ell} \rho^\ell) \\
\leq O(\frac{\sigma\sqrt{\log n}}{\sqrt{n}} \cdot (1+\rho)^k) \\
\leq O(\frac{\sigma\sqrt{\log n}}{\sqrt{n}}) \quad \text{by Lemma A.6} \\
\leq O(\frac{1}{4\sqrt{n}})
\end{aligned}
$$

Where the last inequality follows assuming $\sigma \leq \frac{1}{C'\sqrt{\log n}}$ for some large enough constant $C'$. Finally, to bound the $k^{th}$ order term, we have that with high probability, $\eta^{-k}|e_u^\top R'^k g| \leq (\frac{C_1}{\eta\sqrt{np}})^k \sigma\sqrt{\log n}$.

Putting everything together, and using the fact that $\eta \geq \gamma(1 - o(1))$ with high probability, we have that for all $u \in V$,

$$|e_u^\top s - \frac{1}{\eta^k} \tilde{A}^k x| \leq \frac{1}{\eta^k}(|e_u^\top(\tilde{A}^k - ss^\top)s| + |e_u^\top \tilde{A}^k g|) \leq \frac{1}{2\sqrt{n}} + \left(\frac{C}{\gamma\sqrt{np}}\right)^k \sigma\sqrt{\log n}$$

for some absolute constant $C$ $\qquad\square$

Given Proposition 7.1, we can derive Theorem 4.2 by using the standard reduction in Lemma 5.1.

*Proof.* (Theorem 4.2) By applying Lemma 5.1, we see that given $m$-dimensional features with feature matrix $X$, we can transform it to a centered 1-dimensional feature vector $x = Xw + b = s + g$ where $g \sim \mathcal{N}(0, \sigma'^2 I)$ and $\sigma' = \frac{4\sigma}{\sqrt{n}\|\nu - \mu\|}$. Thus, we have $\|\nu - \mu\| = \frac{4\sigma}{\sigma'\sqrt{n}}$. By Proposition 7.1, our 1-dimensional features become linearly separable as long as $\left(\frac{C}{\gamma\sqrt{np}}\right)^k \sigma'\sqrt{\log n} < \frac{1}{2\sqrt{n}}$ for some absolute constant $C$. Given our expression of $\sigma'$ in terms of the mean distance, this is equivalent to

$$\|\nu - \mu\| \geq \sigma\left(\frac{C}{\gamma\sqrt{np}}\right)^k \sqrt{\log n}$$

In Proposition 7.1, we also needed to assumed that $\sigma' \leq O(\frac{1}{\sqrt{\log n}})$, which implies that

$$\|\nu - \mu\| \geq \Omega(\sigma\sqrt{\frac{\log n}{n}})$$

$\qquad\square$

## D.2 Message Passing Error Bound

In this section, we will prove our main technical lemma, Lemma D.5. By Corollary D.4, it suffices to control the term $|e_u^\top R^k s|$ in order to control $|e_u^\top R'^k s|$. Since this is the sum over many dependent random variables, we cannot easily compute its moment generating function. Instead, we will compute the moments directly. In particular, we will apply Markov's inequality using the $2t^{th}$ moment of this random variable $\mathbb{E}[(e_u^\top R^k s)^{2t}]$ for an appropriately chosen $t$. We now state our main result for this section as follows:

**Proposition D.6.** *Suppose $R$ is an $n \times n$ symmetric random matrix where for each $1 \leq i \leq j$, $R_{i,j} = 1 - p_{i,j}$ with probability $p_{i,j}$ and $-p_{i,j}$ with probability $1 - p_{i,j}$ and are independent. Let $p = \max_{i,j} p_{i,j}$, and suppose $p \geq \Omega(\frac{\log^3 n}{n})$. Then we have*

$$\mathbf{Pr}\left[|e_u^\top R^k s| > \frac{1}{\sqrt{n}}(C\sqrt{np\log n})^k\right] \leq \exp(-\Omega(C\log n))$$

*Proof.* The term $e_u^\top R^k s$ can be written as the sum over all walks of length $k$ originating from $u$. Let $\mathcal{W}_{u,k}$ denote the set of all walks of length $k$ originating at $u$. For brevity, we will use $\mathcal{W}$ to denote $\mathcal{W}_{u,k}$. For each $w \in \mathcal{W}$, let $w(j)$ be the $j^{th}$ vertex in the walk. Note that $w(0) = u$ always. Then we have

$$e_u^\top R^k s = \sum_{w \in \mathcal{W}} \prod_{j=1}^{k} R_{w(j-1),w(j)} s(w(k))$$

Thus, we have

$$\mathbb{E}[(e_u^\top R^k s)^{2t}] = \sum_{w_1, w_2 \ldots w_{2t} \in \mathcal{W}} \mathbb{E}[\prod_{i=1}^{2t} \prod_{j=1}^{k} R_{w_i(j-1),w_i(j)} s(w_i(k))]$$

$$\leq \sum_{w_1, w_2 \ldots w_{2t} \in \mathcal{W}} |\mathbb{E}[\prod_{i=1}^{2t} \prod_{j=1}^{k} R_{w_i(j-1),w_i(j)}]| \cdot \prod_{i=1}^{2t} |s(w_i(k))|$$

$$= \frac{1}{n^t} \sum_{w_1, w_2 \ldots w_{2t} \in \mathcal{W}} |\mathbb{E}[\prod_{i=1}^{2t} \prod_{j=1}^{k} R_{w_i(j-1),w_i(j)}]|$$

Where the last equality follows from the fact that $|s(v)| = 1/\sqrt{n}$ for all $v \in V$. Now, for each $\vec{w} = (w_1, ... w_{2t}) \in \mathcal{W}^{2t}$ and edge $h \in n \times n$, let $\#_{\vec{w}}(h)$ be the number of times the edge $h$ occurs in the graph $w_1 \cup w_2 \cup, ... w_{2t}$. In other words, for $h = \{a, b\}$, $\#_{\vec{w}}(h)$ is the number of times the variable $R_{a,b}$ occurs in the product $\prod_{i=1}^{2t} \prod_{j=1}^{k} R_{w_i(j-1), w_i(j)}$. We will also let $|\vec{w}|$ denote the number of unique edges in the graph formed by the union of these $2t$ walks. For example, if $\vec{w}$ consist of the walks $(1, 2, 3)$, and $(1, 2, 4)$, then we have $\#_{\vec{w}}(1, 2) = 2$, $\#_{\vec{w}}(2, 3) = 1$, $\#_{\vec{w}}(2, 4) = 1$, and $|\vec{w}| = 3$. Using this notation, we have

$$\sum_{w_1, w_2 ... w_{2t} \in \mathcal{W}} |\mathbb{E}[\prod_{i=1}^{2t} \prod_{j=1}^{k} R_{w_i(j-1), w_i(j)}]| = \sum_{\vec{w} \in \mathcal{W}^{2t}} \prod_{h \in [n] \times [n]} |\mathbb{E}[R_h^{\#_{\vec{w}}(h)}]|$$

Now we note that each $R_h$ has $\mathbb{E}[R_h] = 0$ and for all $k \geq 2$, we have

$$|\mathbb{E}[R_h^k]| \leq \mathbb{E}[|R_h^k|] = p_h(1 - p_h)^k + p_h^k(1 - p_h) = p_h((1 - p_h)^k + p_h^{k-1}(1 - p_h)) \leq p_h \leq p$$

Thus, the term $\prod_{h \in [n] \times [n]} |\mathbb{E}[R_h^{\#_{\vec{w}}(h)}]|$ is only nonzero if in the union of the edges in the walks $w_1, ... w_{2t}$, each edge is counted at least twice, and if it is non-zero, then it is the product of at most $|\vec{w}|$ terms of value at most $p$. We now let $\mathcal{W}_{pair}^{2t}$ be the set of such walks, which we will denote as "valid walks". As an example, when $t = 2$ and $k = 2$, the walks $\{(1, 2, 3), (1, 2, 3)\}$ would be valid but $\{(1, 2, 3), (1, 2, 4)\}$ would not be valid. Now we have

$$\sum_{\vec{w} \in \mathcal{W}^{2t}} \prod_{h \in [n] \times [n]} |\mathbb{E}[R_h^{\#_{\vec{w}}(h)}]| \leq \sum_{w \in \mathcal{W}_{pair}^{2t}} p^{|\vec{w}|}$$

$$= \sum_{\ell=1}^{tk} p^\ell |\{\vec{w} \in \mathcal{W}_{pair}^{2t}, |\vec{w}| = \ell\}|$$

Now, what we have left is a counting problem. We need to count the number of valid sets of walks whose union has exactly $\ell$ distinct edges. We note that $\ell$ can be at most $tk$ because otherwise, there must be an edge that is counted at most once, making the walks invalid. Since it is difficult to count this quantity exactly, we will just upper-bound it as follows:

**Proposition D.7.** *Suppose $p \geq \frac{\log^3 n}{n}$ and $t \leq \frac{\log n}{2k}$. Then we have:*

$$|\{\vec{w} \in \mathcal{W}_{pair}^{2t}, |\vec{w}| = \ell\}| \leq \binom{2tk}{2\ell}(2\ell - 1)!! \cdot \ell^{2kt - 2\ell}(n)^\ell$$

The notation of $n!!$ denotes $n \cdot n - 2 \cdot n - 4 ... 1$. Given this upper bound on the number of valid sets of walks, we are ready to give our final bound. We will let $t = \frac{C_1 \log n}{2k}$ for large enough constant $C_1$. Note that the only constraint on $t$ is that it is a positive integer, and this is feasible as long as $k = O(\log n)$. Since $p \geq \frac{\log^3 n}{n}$, we have $np \geq (tk/C_1)^3$. We will make the substitution $b = tk - \ell$. Then we have

$$\sum_{\ell=1}^{tk} p^\ell |\{\vec{w} \in \mathcal{W}_{pair}^{2t}, |\vec{w}| = \ell\}| \leq \sum_{b=0}^{tk-1} (np)^{tk-b}(tk - b)^{2b}(2tk - 2b - 1)!! \binom{2tk}{2b}$$

$$\leq \sum_{b=0}^{tk-1} (np)^{tk-b}(2tk)^{tk-b}(2tk)^{2b} \frac{(2tk)^{2b} e^{2b}}{(2b)^{2b}}$$

$$\leq (np)^{tk}(2tk)^{tk} \sum_{b=0}^{tk-1} \frac{(2tk)^{3b} e^{2b}}{(2b)!(np)^b}$$

$$\leq (np)^{tk}(2tk)^{tk} \sum_{b \geq 0} \frac{(C_1 e^2)^b}{(2b)!}$$

$$\leq O((np)^{tk}(2tk)^{tk})$$

In the second line, we used the fact that $(2tk - 2b - 1)!!$ is the product of $tk - b$ terms of value at most $2tk$ as well as the standard upper bound of $\binom{2tk}{2b} \leq \frac{(2tk)^{2b} e^{2b}}{(2b)^{2b}}$, the second last bound follows

from our assumption on $p$, and the final bound follows from the fact that the sum converges. Putting all of this together, we have that for $t = \frac{C_1 \log n}{2k}$ and $p \geq \frac{\log^3 n}{n}$,

$$\mathbb{E}[(e_u^\top R^k s)^{2t}] \leq O(\frac{(2tknp)^{tk}}{n^t})$$

Now, we can apply Markov's inequality on the $t^{th}$ moment with $t = \frac{C_1 \log n}{2k}$

$$\mathbf{Pr}\left[[] \, |e_u^\top R^k s| > \frac{1}{\sqrt{n}}(C\sqrt{np \log n})^k\right] \leq \frac{n^t \mathbb{E}[(e_u^\top R^k s)^{2t}]}{C^{2tk}(np \log n)^{tk}}$$
$$\leq O(\frac{(np \log n)^{tk} C_1^{tk}}{C^{2tk}(np \log n)^{tk}})$$
$$\leq O(\frac{\sqrt{C_1}^{-\log n}}{C^{\log n}})$$
$$= \exp(-\Omega(\log n))$$

for a sufficiently large constant $C$ □

Now, we combine our results from degree-concentration and the message passing error bound to prove the main result of this section, Lemma D.5.

*Proof.* (Lemma D.5) First, note that for $k = 0$, the bound clearly holds, as $|e_u^\top s| = \frac{1}{\sqrt{n}}$. Now for general $k$, we start by applying Corollary D.4 to obtain

$$|e_u^\top R'^k s| \leq \frac{1}{d^k}|e_u^\top R^k s| + O(\frac{kC_1^k \sqrt{\log n}}{\sqrt{np}^k \cdot \sqrt{n}})$$

For some constant $C_1$. Now by Proposition D.6, we have that with high probability $e_u^\top R^k s \leq \frac{1}{\sqrt{n}}(C_2\sqrt{np \log n})^k$ for some constant $C_2$ and $d \geq \frac{1}{2}(p + q)n(1 - o(1))$. Thus, we have $\frac{1}{d^k}|e_u^\top R^k s| \leq \frac{1}{\sqrt{n}}\left(C_2\sqrt{\frac{\log n}{pn}}\right)^k$. Note that $\sqrt{\log n}^k$ is bigger than $k\sqrt{\log n}$ for large enough $n$. Thus, the $\frac{1}{d^k}|e_u^\top R^k s|$ term clearly dominates so we can take $C$ to be around $\max(C_1, C_2)$ to obtain our upper bound. Finally, by applying union bound, we can ensure that with high probability, bound holds for all $u \in V$. □

### D.2.1 Proof of proposition Proposition D.7

In this section, we complete our proof of the main message passing error bound by proving the upper bound on the number of valid sets of walks of length $k$.

*Proof.* (Proposition D.7) The idea to bound the quantity $|\{\vec{w} \in \mathcal{W}_{pair}^{2t}, \ |\vec{w}| = \ell\}|$ is to first count all the ways to partition the edges of the $2t$ walks into exactly $\ell$ groups of size at least 2. Then, for each partition, we count the number of ways to assign the vertices of the walks in a way such that all edges of the same group are assigned to the same pair of vertices. In the grouping stage, we first pick $2\ell$ walk edges and pair them with each other, assigning a new group to each pair. Then for the rest of the $2tk - 2\ell$ edges, we assign each of them to one of the $\ell$ groups. This ensures that each group has at least two edges. The number of ways to pick $2\ell$ edges is $\binom{2kt}{2\ell}$. The number of ways to pair these edges is $(2\ell - 1)!! = (2\ell - 1)(2\ell - 3)... \cdot 1$. Finally, the number of way to assign the rest of the edges to one of the groups is $\ell^{(2kt-2\ell)}$. Thus, the total number of valid partitions of the edges is at most $\binom{2tk}{2\ell}(2\ell - 1)!! \cdot \ell^{2kt-2\ell}$. Note that this upper bound is tight when $\ell = tk$ but becomes less tight as $\ell$ becomes smaller.

Given a partition of the edges, we now count the number of ways to assign the vertex variables $w_i(j)$. We will give a simple assignment algorithm such that every valid assignment is a possible output of the algorithm. Note that not every output of the algorithm is necessarily valid, so we are actually over-counting.

To see that every possible valid set of walks can be outputted by this procedure, if we are given some $\vec{w} = (w_1, ... w_{2t}) \in \mathcal{W}_{pair}^{2t}$, we can take the partitioning of the edges in the first step to be to be the partition induced by the union of the edges of the $2t$ walks. Then, we can simply follow the vertices of the walks in the same order as in our assignment procedure and see that each time we encounter a new edge corresponds to the first case of the inner for-loop and each time we traverse an already traversed edge corresponds to the second case in the inner for-loop where the procedure does not fail. Thus, the number of valid sets of walks with $\ell$ distinct edges is indeed upper bounded by the total number of possible outputs to our vertex assignment procedure.

To bound the number of possible outcomes in our procedure, we see that in first case of the inner for-loop, there are at most $n$ possible assignments for the vertex $w_i(j)$. In the second step, there is only one possible assignment if the procedure doesn't fail. Since we encounter the first case of the for-loop at most $\ell$ times, the total number of vertex assignments given a fixed partitioning of edges is at most $n^\ell$. Finally, this gives us the desired bound:

$$|\{\vec{w} \in \mathcal{W}_{pair}^{2t}, \ |\vec{w}| = \ell\}| \leq \binom{2tk}{2\ell}(2\ell - 1)!! \cdot \ell^{2kt - 2\ell}(n)^\ell$$

$\square$

# E   Proofs for Section 8

In this section, we will prove our main results for multi-class analysis, Theorem 8.1. First, we will characterize the convolution matrix in terms of the expected adjacency matrix:

**Lemma E.1.** *In the multi-class setting, the convolution matrix, $\tilde{A}$, can be decomposed as $\tilde{A} = M + R'$ where:*

- *$M$ has rank $L-1$, with $L-1$ eigenvalues equalled to $\frac{(p-q)n}{dL} = \lambda$. Also, $MU = \lambda U$*

- *$R'$ is a random matrix such that with probability at least $1 - n^{-\Omega(1)}$, $\|R'\| \leq C(\frac{1}{d}(\sqrt{np(1-p)/L} + \sqrt{nq(1-q)})) = \delta$*

*Proof.* The expected adjacency matrix in the multi-class setting can be written as

$$\mathbb{E}[A] = \begin{bmatrix} pJ_{n/L} & qJ_{n/L} & \dots qJ_{n/L} \\ qJ_{n/L} & pJ_{n/L} & \dots qJ_{n/L} \\ qJ_{n/L} & \dots & \dots qJ_{n/L} \\ qJ_{n/L} & qJ_{n/L} & \dots pJ_{n/L} \end{bmatrix} = (qJ_L + (p-q)I_L) \otimes J_{n/L}$$

The top (normalized) eigenvector of $\mathbb{E}[A]$ is $\frac{1}{\sqrt{n}}\mathbb{1}$. The top eigenvalue is the expected degree $d = pn/L + (L-1)qn/L$. Moreover, $\mathbb{E}[A]$ has rank $L$ and its eigenvectors are of the form $\sqrt{\frac{L}{n}}v \otimes \mathbb{1}_{n/L}$ where $v$ is an eigenvector of the matrix $(qJ_L + (p-q)I_L)$. Since the top eigenvector is $\frac{1}{\sqrt{n}}\mathbb{1} = \frac{1}{\sqrt{L}}\mathbb{1}_L \otimes \sqrt{\frac{L}{n}}\mathbb{1}_{n/L}$, the rest of the eigenvectors of $\mathbb{E}[A]$ must be of the form $\sqrt{\frac{L}{n}}v_2 \otimes \mathbb{1}_{n/L}, \dots \sqrt{\frac{L}{n}}v_L \otimes \mathbb{1}_{n/L}$, where $v_2, \dots v_L$ are an orthonormal basis of the subspace in $\mathbb{R}^L$ orthogonal to $\mathbb{1}_L$. Thus, for $l = 2, ..L$, we have $\mathbb{E}[A](v_l \otimes \mathbb{1}) = \frac{1}{L}(p-q)n(v_l \otimes \mathbb{1})$, which means the second to $L^{th}$ eigenvalues of $\mathbb{E}[A]$ are equalled to $\lambda d$.

Now let $R = A - \mathbb{E}[A]$ and . Then we have

$$\tilde{A} = \frac{1}{\bar{d}}A - \frac{1}{n}\mathbb{1}\mathbb{1}^\top = (\frac{1}{\bar{d}} - \frac{1}{d})A + \frac{1}{d}R + \frac{1}{d}\mathbb{E}[A] - \frac{1}{n}\mathbb{1}\mathbb{1}^\top$$

Now, let $M := \frac{1}{d}\mathbb{E}[A] - \frac{1}{n}\mathbb{1}\mathbb{1}^\top$. Note that the non-zero eigenspace of $M$ is equivalent to 2nd to $L^{th}$ eigenspace of $\mathbb{E}[A]$ and the corresponding eigenvalue is $\lambda$. To show that $MU = U$, let $u$ be any column of $U$. By our assumption that each class has one center, the entries of $u$ are constant on each class. If we define $v \in \mathbb{R}^L$ such that $v(l) = u(i)$ for $i \in \mathcal{C}_l$, then we can write $u = v \otimes \mathbb{1}_{n/L}$. Since by our assumption, $u \perp \mathbb{1}_n$, we must have $v \perp \mathbb{1}_L$. Thus, we see that $u$ is exactly in the subspace spanned by the non-zero eigenvectors of $M$, which means $Mu = \lambda u$.

Now, we simply have to bound the spectral norm of the matrix $R' := \frac{1}{d}R + (\frac{1}{\bar{d}} - \frac{1}{d})A$. We start with bounding $\|R\|$. To do so, we can write $R$ as $R = R_p + R_q$, where $R_p$ and $R_q$ contain the entries of $R$ corresponding to intra- and inter- class edges respectively. Note that $R_p$ is block diagonal with $L$ blocks of size $n/L$. Each block is size $n/L \times n/L$ and have i.i.d entries. By Theorem A.4, each block has spectral norm at most $O(\sqrt{np(1-p)/L})$ with probability at least $1 - n^{-\Omega(1)}$. Similarly, we have $\|R_q\| \leq O(\sqrt{nq(1-q)})$ with probability at least $1 - n^{-\Omega(1)}$. This means $\|R\| \leq O(\sqrt{np(1-p)/L} + \sqrt{nq(1-q)})$.

Now, we bound the degree deviation. Recall that $d = \frac{2|E|}{n}$ where $|E|$ is the number of edges in $G$. By applying Theorem A.3 with $t = \sqrt{np(1-p)/L} + \sqrt{nq(1-q)}$, we have

$$\mathbf{Pr}\left[|\bar{d} - d| > t\right] \leq \exp(-\Omega(\frac{t^2}{\max(p,q)})) \leq n^{-\Omega(1)}$$

since $\min(p,q) \geq \log^2 n/n$. Assuming that $|\bar{d} - d| \leq t$ and noting that $\|A\| \leq \bar{d}$, we have $\left\|(\frac{1}{\bar{d}} - \frac{1}{d})A\right\| \leq \|R\| \leq t/d$. Thus, we have $\tilde{A} = M + R'$, where $\|R'\| \leq \delta$ with high probability. $\square$

We will also need to bound the operator norm distance between the $k^{th}$ convolution and $M^k$

**Lemma E.2.** *Suppose $|\lambda| > 2k\delta$. Then with high probability, we have*

$$\left\| \frac{1}{\lambda^k}(\tilde{A}^k - M^k) \right\| \leq 2k\delta/|\lambda|.$$

*Proof.* By Lemma E.1, we have

$$\tilde{A}^k = (M + R')^k = M^k + \sum_{l=1}^{k} \sum_{b \in \binom{[k]}{l}} \prod_{i=1}^{k} M^{1-b(i)} R'^{b(i)},$$

where the inner sum is over bit-strings $b$ of length $k$ with exactly $l$ 1's and $k - l$ 0's. Note that $\|M\| = |\lambda|$ and $\|R'\| \leq \delta$ with high probability. Using the fact that $\|AB\| \leq \|A\|\|B\|$ and triangle inequality, we have

$$\left\| \frac{1}{\lambda^k}((M + R')^k - M^k) \right\| \leq \frac{1}{|\lambda|^k} \sum_{l=1}^{k} \binom{k}{l} \|M\|^{k-l} \|R'\|^l \leq \sum_{l=1}^{k} \binom{k}{l} (\frac{\delta}{|\lambda|})^l = (1 + \frac{\delta}{|\lambda|})^k - 1$$

Our assumption that $|\lambda| > 4\delta k$ implies the RHS is at most $2k\delta/|\lambda|$ by Lemma A.6. $\qquad\square$

Now we are ready to prove Theorem 8.1.

*Proof.* We can express the total squared error after $k$ convolutions as:

$$\sum_{i=1}^{n} \left\| x_i^{(k)} - \mu_i \right\|^2 = \left\| X^{(k)} - U \right\|_F^2$$

We decompose our data as $X^{(k)} = U + G$, where $G$ is a Gaussian matrix with i.i.d $N(0, \sigma^2)$ entries. Recall that we take our scaling factor to be $1/\lambda^k$. Thus, we have

$$\left\| X^{(k)} - U \right\|_F^2 = \left\| \frac{1}{\lambda^k} \tilde{A}^k(U + G) - U \right\|_F^2 \leq 2 \left\| \frac{1}{\lambda^k} \tilde{A}^k U - U \right\|_F^2 + 2 \left\| \frac{1}{\lambda^k} \tilde{A}^k G \right\|_F^2.$$

By Lemma E.1, we have $MU = \lambda U$. Let $u_1, ... u_m$ be the columns of $U$. Then, we have

$$\begin{aligned}
\left\| \frac{1}{\lambda^k} \tilde{A}^k U - U \right\|_F^2 &= \left\| \frac{1}{\lambda^k}(\tilde{A}^k - M^k)U \right\|_F^2 \\
&= \sum_{i=1}^{m} \left\| \frac{1}{\lambda^k}(\tilde{A}^k - M^k)u_i \right\|^2 \\
&\leq \sum_{i=1}^{m} \left\| \frac{1}{\lambda^k}(\tilde{A}^k - M^k) \right\|^2 \|u_i\|^2 \\
&= \left\| \frac{1}{\lambda^k}(\tilde{A}^k - M^k) \right\|^2 \|U\|_F^2 \\
&\leq O(k\delta/|\lambda|)^2 \|U\|_F^2
\end{aligned}$$

where the last inequality follows from Lemma E.2. By Lemma A.5, we have, with high probability,

$$\left\| \frac{1}{\lambda^k} \tilde{A}^k G \right\|_F^2 \leq \frac{1}{\lambda^{2k}} Tr(\tilde{A}^{2k}) \sigma^2 m \log n$$

Since adding $R'$ to $M$ perturbs its eigenvalues by at most $\delta$ (Theorem A.1), we have

$$\frac{1}{\lambda^{2k}} Tr(\tilde{A}^{2k}) \leq (1 + \delta/|\lambda|)^{2k}(L - 1) + n(\delta/|\lambda|)^{2k}$$

$|\lambda| > 4\delta k$ implies $(1 + \delta/|\lambda|)^{2k} \leq O(1)$. Now let $n_e$ be the number of points $i$ such that $\left\| x_i^{(k)} - \mu_i \right\| \geq \Delta/2$. Then we have

$$n_e \Delta^2 \leq 4 \left\| X^{(k)} - U \right\|_F^2 \leq O((k\delta/|\lambda|)^2 \|U\|_F^2 + (L + n(\delta/|\lambda|)^{2k})m\sigma^2 \log n)$$

Dividing both sides by $\Delta^2$ gives us the desired bound.

Finally, we recall that $\Delta$ is the minimum distance between centers. Thus if $\left\| x_i^{(k)} - \mu_i \right\| < \Delta/2$, then it is closer to its own center than any other center. Thus, the softmax classifier can correctly classify all such points. $\qquad\square$

### E.1 Additional Figures for Multi-class simulation

Finally, we conclude our section with some additional figures to illustrate the performance of the corrected convolution on synthetic data. We compare the corrected and uncorrected convolutions via both linear and non-linear models. Our means are class means are given by the standard basis vectors. For the linear model (figs 4a – 4c), we look at five 1-vs-all classifiers followed by a softmax to predict the class label, while the non-linear method (figs 4d – 4f) follows a typical two-layer MLP-based architecture. In both cases, we observe that the corrected convolutions do not deteriorate in performance as the number of convolutions increases. Note that while the performances are similar between linear and non-linear classification, non-linear classification is required in general when each individual class mean cannot be separated from all others via a linear hyperplane.

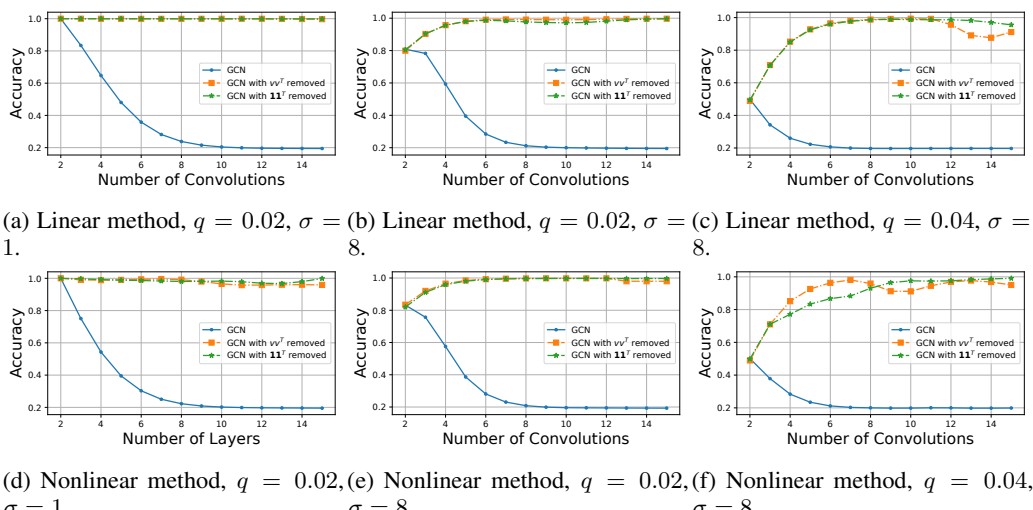

(a) Linear method, $q = 0.02, \sigma = 1$.

(b) Linear method, $q = 0.02, \sigma = 8$.

(c) Linear method, $q = 0.04, \sigma = 8$.

(d) Nonlinear method, $q = 0.02, \sigma = 1$.

(e) Nonlinear method, $q = 0.02, \sigma = 8$.

(f) Nonlinear method, $q = 0.04, \sigma = 8$.

Figure 4: Accuracy plot (averaged over 50 trials) on CSBM data with 5 balanced classes, 500 nodes per class and orthogonal means, with fixed $p = 0.1$.

