# OpenReview forum: "Analysis of Corrected Graph Convolutions"
_NeurIPS.cc/2024/Conference — NeurIPS 2024 poster_

### Official Review · Reviewer_kcVu · 2024-06-19

**Soundness:** 2
**Presentation:** 1
**Contribution:** 3
**Rating:** 6
**Confidence:** 4

**Summary:**

This paper studies the effects of removing the top eigenvector of the adjacency matrix used for aggregation in graph convolutions. For the contextual stochastic block model (CSBM), this is theoretically shown to be beneficial. The authors provide several theoretical statements describing misclassification ratios and chances and chances of achieving linear separation. Experiments confirm the superiority of removing the smooth eigenvector for the CSBM.

**Strengths:**

* The background, literature review, and preliminaries are nicely presented.
* The ideas behind the proofs are interesting.
* The conducted experiments confirm the claimed benefits of removing the top eigenvector for CSBM.

**Weaknesses:**

* The structure of the paper is confusing. In Section 4, two Theorems are provided without proofs. Proofs are also not provided in the Appendix. Then, there is a proof sketch for the Theorems in Section 6.
* A clear focus would help the accessibility of this work. It would suffice to consider either $\hat{A}$ or $\tilde{A}$ and Theorem 4.1 or Theorem 4.2 while providing more details for the selected case.
* Theoretical statements and their implications are hard to understand. Clearly stating all symbols and providing additional details would help.
* The implications of this work are not clear to me. We know that $M^kx$ gets dominated by the top eigenvector of M with rate $\lambda_2/\lambda_1$. Setting the largest eigenvalue to zero, results in dominance of the second eigenvector. As pointed out in this work, this eigenvector corresponds to sparse and balanced bipartitions. This analysis seems like a complicated way to come to the same conclusion.
* The practical usage of removing the top eigenvector seems very limited, as one dominating signal is exchanged for another. This is confirmed in Figure 3.

**Questions:**

* What are the benefits of this analysis against existing insights that the signal corresponding to the top eigenvector of a matrix gets amplified the most and dominates representations?
* What are potential practical implications that can now be developed?
* l. 332: Why is the O-notation used for $p=O(log^3 n/n)$?

**Limitations:**

Limitations are not explicitly presented.

---

> ### Author Rebuttal · Authors · 2024-08-07
>
> We answer the concerns below.
>
> > The structure of the paper is confusing. Two Theorems are provided without proofs. Proofs are also not in the Appendix.
>
> We prove both our theorems rigorously in the appendix. For the proof of Theorem 4.1, we refer to Appendix C (Title: Proofs in Section 6). Please check line 611, where we start by proving the essential lemmas, followed by the proof of Theorem 4.1 in line 662. For Theorem 4.2, the sketch is in section 7 (line 277) and the full proof is in Appendix D titled `Proofs in Section 7 (line 676)`. The explicit proof of Theorem 4.2 is given in `line 760`. We will point to the proofs explicitly with hyperlinks in the main paper in our revision.
>
> > It would suffice to consider either $\hat{A}$ or $\tilde{A}$ and Theorem 4.1 or Theorem 4.2.
>
> Thank you for the suggestion. We believe that both Theorem 4.1 and 4.2 are important. In community detection results for the SBM, it is customary to study both exact and partial recovery results (see \[E. Abbe, Community Detection and Stochastic Block Models, JMLR 2018\]). In our paper, we provide answers to such customary questions for node classification for the CSBM, where we also have node feature information in addition to the graph.
>
> > Theoretical statements and their implications are hard to understand. Clearly stating all symbols and providing additional details would help.
>
> We provide an extensive discussion on the theorems and their meaning in terms of the number of convolutions and the SNR in the data (see lines 170 to 188 for Theorem 4.1, and lines 193 to 202 for Theorem 4.2). It would help if the reviewer could point us to the part that is hard to understand so we can explain it better and modify our paper accordingly. It would be helpful if the reviewer could tell us which symbols and additional details are required.
>
> > The implications of this work are not clear to me. Setting the largest eigenvalue to zero results in dominance of the 2nd eigenvector, corresponding to sparse and balanced bipartitions. This analysis seems like a complicated way to come to the same conclusion.
>
> We respectfully disagree with the reviewer's perspective. There is a large body of literature in the GNN community where people care about precisely quantifying the classification results and non-asymptotic behavior of these models. For example, \[ref line 443, Keriven: Not Too Little, Not Too Much\] studied classification guarantees of graph convolution in simplified statistical models and showed improved classification guarantees compared to no convolutions. However, the analysis was limited to at most $2$ convolutions. In \[ref line 494, Wu et al.\] the authors studied the non-asymptotic behavior of graph convolutions in the CSBM and precisely characterized how many convolutions it takes before the oversmoothing effect overtakes the aggregating effect of graph convolution. However, the paper does not give classification guarantees of the model. In our paper, we analyze the corrected graph convolution in the CSBM, which, in this model, mitigates the oversmoothing effect. As a result, we can give partial and exact recovery guarantees for up to $O(\log{n})$ convolutions, which has not been done in all the aforementioned works. We also precisely quantify the number of convolutions it takes to obtain our recovery results. We also note that the proof of our exact recovery result (Theorem 4.2) requires much more sophisticated techniques than simply analyzing the rate of convergence of the eigenvalues to $\lambda_1$. This is because we have to bound the distance $\|A^kx - s\|_\infty$ instead of $\|A^kx - s\|_2$, which means simple spectral analysis is insufficient. To overcome this, we used careful analysis of the moment of each entry of our convolved feature vector.
>
> Finally, we argue that the intuition behind our analysis being simple is a good thing because it makes it more accessible to the general audience. Our analysis formalizes this intuition that most people can interpret into concrete recovery guarantees whereas previous studies did not.
>
>
>
> > The practical usage of removing the top eigenvector seems very limited, as one dominating signal is exchanged for another. This is confirmed in Fig 3.
>
> The eigenvector which we remove has no information about the class membership. The second eigenvector which dominates after removing the first has useful classification information. Therefore, simply stating that exchanging one signal for another is not useful, is not correct. We prove this in our Theorems 4.1 and 4.2. Fig 3 does not confirm the reviewer's statement. In fact, Fig 3 shows that removing the top eigenvector allows the model to have better performance as the number of convolutions increases. As we point out in lines 348 to 350, the idea of removing the top eigenvector is helpful in practice and has been implemented in widely cited studies (see \[ref line 512, Zhao and Akoglu\]).
>
> > What are the benefits of this analysis against existing insights that the signal corresponding to the top eigenvector of a matrix gets amplified the most?
>
> We have answered this question in detail in our reply about Weakness 4 above.
>
> > What are potential practical implications?
>
> This is similar to weakness 5 so see the response there. To re-iterate, based on our analysis, we could recommend removing the top eigenvector from the convolution matrix. As we point out in lines 348 to 350, this is helpful in practice and has been implemented in widely cited studies (see \[ref line 512, Zhao and Akoglu\]).
>
> > l. 332: Why is the O-notation used?
>
> We will update the precise value of $p$ in the experiments. We used $O$ notation to denote that there is a constant factor associated with the value.
>
> If the reviewer has more actionable suggestions to further improve our work, please kindly let us know. We would be happy to address any further questions. We hope the clarifications are sufficient for them to consider raising their score.

---

> ### Comment · Reviewer_kcVu · 2024-08-11
>
> I thank the reviewers for their detailed feedback. I can now better understand why this work can be interesting from a mathematical and statistical point.
>
> I would like to hear more details about the following:
> * In the referenced work [ref line 443, Keriven: Not Too Little, Not Too Much], the author shows that some rounds of aggregation can be helpful while too many iterations are hurtful for their considered task. Here, it seems like more rounds of aggregation are always beneficial, but fewer rounds can be sufficient. Can this study be extended in the future to include cases where the task does not align so well with the dominating eigenvector? For Figure 3, performance seems to be monotonously decreasing, so (zero or one?) iterations seem to be optimal. What would tasks look like for which some iterations of your corrected convolutions are beneficial while the performance in the limit becomes worse?
>
> Two additional comments on terms that I found irritating but not critical:
> * Convolution: There is no definition of what convolution means. It seems like it's just the aggregation part $\mathbf{A}$ and powers of it. Referring to this as convolution is confusing to me, as no filter is considered. In the referenced work above, the author calls this $k$ rounds of aggregation, which would also make the statements in this work clearer.
> * Corrected convolution: Was this term used previously in the literature? What makes this aggregation more "correct"? I agree that it seems more correct for the tasks considered in this work. However, I would argue that depending on the graph and task, various aggregation matrices can be more "correct".

---

> > ### Comment · Reviewer_kcVu · 2024-08-13
> >
> > The above comment was apparently not visible to the authors. I apologize and hope my question still comes in time.

---

> ### Author Response · Authors · 2024-08-13
>
> We thank the reviewer for their reply. However, we would like to point out that this post was made very close to the deadline and during the authors' nighttime. A proper reply to the reviewer takes several hours. We provide a reply to the best of our ability given the limited time.
>
> We want to re-iterate the fact that the corrected convolution is not ours. We mention this in our first reply above and we point this out in lines 348 to 350. This idea is already implemented in widely cited practical papers [ref line 512 (ICLR 2020)]. We provide a rigorous analysis.
>
> > In the referenced work [ref line 443, Keriven: Not Too Little, Not Too Much], ...  task.
>
> The result that the reviewer claims seems to be Theorem 2. We would like to clarify what is proved in the main theorem of this paper. The authors don't provide a non-asymptotic analysis as we do in our paper. They only provide an existential result. They prove that 1 convolution can be better, under specific assumptions, than 0 convolutions. They also prove that 1 convolution is better than infinite convolutions. This implies that there might exist a $k^* \ge 1$ which is optimal. However, they never rigorously analyze the performance of convolution for $k>1$ and $k<\infty$. In fact, the authors mention in their paper that "In the next section, we also derive an intuitive expression for $R(k)$ (although without rigorous proof), which we observe to match the numerics quite well."
>
> We would like to point out that it is technically extremely challenging, and requires a whole different set of methodology, to achieve non-asymptotic analysis as we do in our paper, and we hope the review appreciates it.
>
> > Can this study be extended ...  in the limit becomes worse?
>
> We will reply to this question by focusing on other tasks within node-classification, since node-classification is the focus of our paper. By dominating eigenvector we assume that the reviewer implies the second or higher eigenvectors. The dominant (first) eigenvector does not hold meaningful classification information, therefore removing it helps. We prove this in our Theorems 4.1 and 4.2. Also, Fig 3 shows that removing the top eigenvector allows the model to have better performance as the number of convolutions increases.
>
> We interpret the reviewer's comment as a question about why the performance decreases for $k$ larger than 1 or 2. The reason is not because we remove the dominant (first) eigenvector. As we point out this is beneficial, since the first eigenvector holds no information about the classes. The reason is that in the particular real data which we used there are eigenvectors with small eigenvalues which might contain meaningful information. Asymptotically, the corrected convolution, which we repeat is not our idea, see comments above, might miss the information from such eigenvectors. Intuitively, this could be the case in a graph where there are small communities within larger communities in the given graph structure. The top eigenvectors are likely to be correlated with the larger communities. To capture the information from small eigenvalues, a different convolution is needed which filters eigenvectors in an appropriate way. However, we would like to point out again, that the first eigenvector will be filtered out in this case too since it's not meaningful.
>
> To analyze such a scenario, we would have to change our random data model. That's because in the current data model the first few top eigenvectors (excluding the first one) are the most meaningful ones, and eigenvectors for small eigenvalues are just noise. Technically, a non-asymptotic spectral analysis could also work in this hypothetical new data model. The corrected convolution in our paper is likely to lose performance as $k$ increases. A different convolution would be needed to better capture the information of eigenvectors of small eigenvalues. This study is indeed interesting, however, we would like to mention that it is far from the goal of our present paper.
>
> We would be happy to mention the above in our paper.
>
> > Convolution: ... in this work clearer.
>
> We will make sure to clarify this in our paper. We come from a graph neural network background where this definition is ubiquitous and we didn't explicitly specify it as a definition.
>
> > Corrected convolution: ... can be more "correct".
>
> We chose the word corrected since in the literature of statistical community detection (no node features), the word is often used for modified versions of the original SBM, for example: B. Karrer and M. E. J. Newman, Stochastic blockmodels and community structure in networks, Phys. Rev. E 83 (2011). In this case, the fact that one does degree "correction" does not necessarily mean that it is a universal correction for any possible problem. It's a name which makes sense within a specific context. We can offer to change the title to "Analysis of Eigen-Corrected Graph Convolutions" if the reviewer believes that this is more specific.

---

> ### Comment · Reviewer_kcVu · 2024-08-13
>
> I want to thank the reviewers for their quick and detailed answers. I am now convinced of this work's benefits to the graph learning community and am open to accepting this paper. I have changed my score to 6.
>
> I want to state my final opinions below, to which the authors do not need to reply:
>
> >The dominant (first) eigenvector does not hold meaningful classification information, therefore removing it helps. [...] Fig 3 shows that removing the top eigenvector allows the model to have better performance as the number of convolutions increases.
>
> The best-achieved performance of the original aggregation and corrected aggregation are quite similar (both for k=1). Stating that the corrected aggregation retains more task-related information would better describe the behavior. Identifying non-SBM tasks for which "removing the top eigenvector allows the model to have better performance as the number of convolutions increases" is actually true, and the max performance for the corrected aggregation is better compared to the standard aggregation, which is still open and interesting for future work.
>
> > We come from a graph neural network background where this definition is ubiquitous and we didn't explicitly specify it as a definition.
>
> In my graph neural network background, a graph convolution consists of a feature transformation and an aggregation part. As this work considers the aggregation part, I would personally find it more precise to call it $k$ rounds of aggregation instead of $k$ rounds of convolution.
>
> >We can offer to change the title to "Analysis of Eigen-Corrected Graph Convolutions" if the reviewer believes that this is more specific.
>
> I don't mind if the authors keep the original title. To me personally, something like "Analysis of CSBM-Corrected Graph Aggregations" would seem clearer.

---

> > ### Author Response · Authors · 2024-08-13
> >
> > We are grateful. Thank you. We will make appropriate modifications to address your concerns in the revised version.

---

### Official Review · Reviewer_uSHp · 2024-07-03

**Soundness:** 3
**Presentation:** 3
**Contribution:** 3
**Rating:** 7
**Confidence:** 3

**Summary:**

This paper studies the concept of oversmoothing via a CSBM modeling of a GNN structure, and views the behaviors of vectors after repeated multiplications of a graph. Importantly, they consider a scheme where a dominant eigenvector is "left out", so that it does not dominate the behavior of the evolution so much, and show enhanced performance over real world datasets.

**Strengths:**

This paper explores an important problem from a simple enough framework that it is tractable. There are a ton of statistical results, which all look reasonable and tell a reasonable story about the balance between learning and oversmoothing.

The idea of discarding the dominant eigenvector is also interesting, and clearly does show improved results in the numerics.

**Weaknesses:**

It's not clear to me how informative / predictive the various statistical results are to oversmoothing in practice. Can there be some figures that show the theoretical bounds, compared with true performance over some simulated examples? For example, one of the theorems uses  Davis/Kahan, which is pretty pessimistic in practice. In general, perhaps the spectral gap alone is not informative enough to showcase the entirety of the oversmoothing behavior, and while that is certainly fine given the complexity of the problem, it still would be useful to see the theorems compared with in practice behavior.

**Questions:**

What is CSBM? I know SBM but what makes it C?

Have you considered scenarios where the data has clear block structure, but it does not 100% overlap with the true labels? will it cause undesirable biases? can those be quantified?

Actually, one interpretation of this is that the adjacency spectrum somewhat mirrors the Laplacian spectrum, but in reverse order. So, the dominant eigenvector corresponds to the nullspace of L, and the second one is the one associated with the mixing parameter in L. So, in that sense, it makes sense why focusing on that eigenspace gives better performance. I would be interested in hearing the authors' view on this.

**Limitations:**

no ethical issues

---

> ### Author Rebuttal · Authors · 2024-08-07
>
> We sincerely thank the reviewer for the insightful questions and comments along with the encouraging review. We answer the questions below.
>
> > Comment/question 1: It's not clear to me how informative / predictive the various statistical results are to oversmoothing in practice. Can there be some figures that show the theoretical bounds, compared with true performance over some simulated examples? For example, one of the theorems uses Davis/Kahan, which is pretty pessimistic in practice. In general, perhaps the spectral gap alone is not informative enough to showcase the entirety of the oversmoothing behavior, and while that is certainly fine given the complexity of the problem, it still would be useful to see the theorems compared with in practice behavior.
>
> We assume that the reviewer would like to see the theoretical classification accuracy bound for Theorems 4.1 and 4.2 in the same plot of our simulated synthetic experiments. Please, let us know if this is not the correct interpretation of your question. We would be happy to update our reply during the discussion period. In case, that's the correct interpretation of your question we provide updated plots with the theoretical bound for partial recovery, Theorem 4.1, in the response pdf. We would like to note that the exact classification thresholds from Theorem 4.2 are demonstrated with vertical lines in the current plots in Figures 1 and 2 of our paper.
>
>
> > Question 2: What is CSBM? I know SBM but what makes it C?
>
> Context is referred to in the literature as the features of the nodes. \textcolor{red}{Are there any historical reasons that the features are called context here? Cite the original CSBM paper and point to Section 3.1}
>
> > Question 3: Have you considered scenarios where the data has clear block structure, but it does not 100\% overlap with the true labels? will it cause undesirable biases? can those be quantified?
>
> This is a very good question. We have thought about this a little bit. If there is a mismatch between the community structure in the graph and classes of the feature vectors of the nodes, then the mismatched vertices will probably be pulled to the opposite means by the convolutions. This means that there will be more misclassified nodes in the partial classification result, and the threshold of exact classification will be worse. We can probably still analyze this case by splitting the feature vector into the part corresponding to correctly matched vertices and the part corresponding to mismatched vertices and analyzing the error from each part separately. If the mismatch is small, we should be able to get reasonable partial classification results. This could be an interesting avenue for future work.
>
> > Question 4: Actually, one interpretation of this is that the adjacency spectrum somewhat mirrors the Laplacian spectrum, but in reverse order. So, the dominant eigenvector corresponds to the null space of L, and the second one is the one associated with the mixing parameter in L. So, in that sense, it makes sense why focusing on that eigenspace gives better performance. I would be interested in hearing the authors' views on this.
>
> We agree with this interpretation, and that's exactly what happens in our analysis for CSBM. We will mention the relation to the Laplacian more explicitly in our revision.
>
> Final remarks
>
> Once again, we would like to express our gratitude for the thorough examination and feedback of our work. We believe that it certainly helped us to improve our manuscript. If the reviewer has more actionable suggestions to further improve our work, please kindly let us know. We would be happy to address any further questions. We hope the clarifications and additional comments are sufficient for them to consider raising their score.

---

> > ### Author Response · Authors · 2024-08-08
> > **Correction of typo in the response**
> >
> > Dear reviewer, we apologize for the typo in the response to Question 2: What makes it C in CSBM?
> > The C (context) here refers to the node features, since the data model consists of node features in addition to the graph.
> > This term has been used in the statistics community to describe such data with a combination of two components: the graph and the node attributes.
> > Please refer to Section 3.1 for a detailed description of the CSBM, and to [Deshpande, Y. and Sen S. and Montanari, A. and Mossel, E. Contextual stochastic block models. Advances in Neural Information Processing Systems, 2018.] where it was introduced.

---

> > ### Comment · Reviewer_uSHp · 2024-08-10
> >
> > Thanks, I am happy with all the responses. The new figures showing the bound vs performance are also very interesting, and a great improvement on the paper. I have no further questions.

---

> > > ### Author Response · Authors · 2024-08-12
> > >
> > > Thank you! We will incorporate the modifications in the revised version of the paper.

---

### Official Review · Reviewer_eGdd · 2024-07-11

**Soundness:** 4
**Presentation:** 3
**Contribution:** 3
**Rating:** 7
**Confidence:** 3

**Summary:**

This paper studies over-smoothing from $k$ rounds of graph convolutions in the Contextual Stochastic Block Model by considering vanilla graph convolutions and a corrected one where the principal eigenvector is removed. Using spectral analysis, the authors derive the partial and exact recovery for both cases and show that the corrected convolution avoids over-smoothing. They derive the classification error for different densities $p$, $q$ and separability $\gamma$ of the underlying graph model.

**Strengths:**

1. The theoretical analysis is thorough and comprehensive. The studied setting is also of practical significance and the authors provide rigorous theoretical analysis.
2. The presentation is clear. The discussions on the theoretical results and assumptions are very helpful in understanding the paper.
3. Experiments also support the theoretical results.

**Weaknesses:**

There are no critical weaknesses in the work. I have a few comments/questions which I list below.
1. One aspect of the result is the dependence on the density of the graph as well as good SNR ratio. Can the authors comment on the need for denser connectivity in the graph for recovery?
2. The performance of the corrected convolution decreases with $k$ in the case of a multi-class setting. The authors reason it as the projection onto the second eigenvector cannot capture the full class information. However, in the balanced data setting, all the eigenvalues of the corrected convolution are the same, with eigenvectors having information about the classes. So, in expectation, shouldn't the performance be unaffected?
3. The analysis considers homophily structure of the graph $p>q$. Can it be extended to heterophily case as well?
4. In the definition of $\hat{A}$ in equation 1, shouldn't the last $D$ be $D^{-1/2}$? The negative sign is missing.

**Questions:**

Please refer to the Weaknesses.

**Limitations:**

The analysis is in linear setting, and extending it to non-linear activations may be challenging.

---

> ### Author Rebuttal · Authors · 2024-08-07
>
> We sincerely thank the reviewer for their thought-provoking questions and comments, and are grateful for the encouraging review. We answer the questions below.
>
> > Question 1: One aspect of the result is the dependence on the density of the graph as well as good SNR ratio. Can the authors comment on the need for denser connectivity in the graph for recovery?
>
> In general, the lower bound on density ($p$ value) is required for degree and matrix concentration (see Proposition 5.3). In case the reviewer is asking why Theorem 4.2 requires a higher density than Theorem 4.1, this is just an artifact of the more technical analysis for Theorem 4.2. We believe with better analysis, we can probably remove the extra $\log n$ requirement in Theorem 4.2. Specifically, the proof of Theorem 4.2 requires some combinatorial counting arguments to bound the moment of the entry-wise error in the convolved feature vector. In counting these combinatorial objects, we sometimes applied some crude upper bounds which are likely not tight.
>
> > Question 2: The performance of the corrected convolution decreases with $k$ in the case of a multi-class setting. The authors reason it as the projection onto the second eigenvector cannot capture the full class information. However, in the balanced data setting, all the eigenvalues of the corrected convolution are the same, with eigenvectors having information about the classes. So, in expectation, shouldn't the performance be unaffected?
>
> That is a good observation. Indeed if the convolution matrix's top eigenvalue has multiplicity $L$, then after many convolutions, we are projecting onto the top $L$-eigenspaces. In response to these insights, we have produced new analysis on multi-class CSBM (see global response), in which we generalize Theorem 4.1 (partial recovery result) to the multi-class setting with balanced classes. If there are $L$ classes, the second eigenvalue of the expected adjacency matrix has multiplicity $L-1$ just as the reviewer pointed out. We show in our analysis that if the perturbation is small and $k$ is not too large, the corrected convolution will still behave like projecting onto the second to $L^{th}$ eigenspace of the expected adjacency matrix.
>
> In the case of real data, it is still likely the case that projecting the features onto the top few eigenspaces is not sufficient to capture all the information of the class labels and could also destroy some relevant information within the feature distribution. However, we still demonstrate that simply removing the top eigenvector leads to improved performance in the oversmoothing regime. We thank the reviewer for this observation and will modify our write-up accordingly to include this discussion.
>
> > Question 3: The analysis considers homophily structure of the graph $p>q$. Can it be extended to heterophily case as well?
>
> Our results hold for the case where $q>p$ as well. See lines $159$ to $161$ in our paper.
>
> > Question 4: In the definition of $\hat{A}$ in equation 1, shouldn't the last $D$ be $D^{-1/2}$? The negative sign is missing.
>
> Thanks for catching the typo. The second term should be normalized by $\mathbf{1}^\top D\mathbf{1}$. Here, $D^{1/2}\mathbf{1}$ is the top eigenvector of the normalized adjacency matrix.
>
> > Limitation: The analysis is in linear setting, and extending it to non-linear activations may be challenging.
>
> In our analysis on binary classification, the linear classifier was sufficient. However, for multi-class data, we could require non-linear classifiers. We have extended our partial recovery result (Theorem 4.1) to the multi-class setting which involve the use of a non-linear classifier (see global response). In our analysis, we have $L$ classes, and we assume features are generated by a Gaussian mixture model with one mean for each class. In our results, we show that the graph signal is measured by the quantity $\lambda = \frac{(p-q)n}{dL}$ where $d$ is the expected degree. The graph noise is bounded by $\delta = O(\frac{1}{d}(\sqrt{np(1-p)/L} + \sqrt{nq(1-q)}))$. We show that as long as $|\lambda| \gtrsim k\delta$, and the cluster means are well separated, then after $k$ rounds of convolution of the data, $1-o(1)$ fraction of points will be closer to the mean of their class than of any other class. Given this guarantee, we can correctly classify $1-o(1)$ nodes using the non-linear classifier: $x\mapsto \text{softmax}(\|x-c_l\|^{2})_{l=1}^L$, where $c_l's$ are cluster means from each class. This is a quadratic classifier, where the means $c_l$ are the learnable parameters.
>
> We note that in our model, we assumed there is only one mean for each class. It is possible to add more complexity to the model by having multiple means in each class and considering different ways they can be distributed. See, for example, the reference in line 385 where the authors analyze an XOR based data model for binary classification, requiring a non-linear classifier. We hope to extend our analysis to capture these more general models and classifiers as well.
>
> Final remarks
>
> Once again, we would like to express our gratitude for the thorough examination and feedback of our work. We believe that it certainly helped us to improve our manuscript. If the reviewer has more actionable suggestions to further improve our work, please kindly let us know. We would be happy to address any further questions. We hope the clarifications and additional comments are sufficient for them to consider raising their score.

---

> > ### Comment · Reviewer_eGdd · 2024-08-09
> >
> > I thank the authors for their clarifications and for providing analysis for multi-class setting. I retain my score as I didn't have any major concerns in my initial review, and I recommend acceptance of the paper.

---

> > > ### Author Response · Authors · 2024-08-10
> > >
> > > We thank the reviewer for recommending acceptance of the paper. We will address all comments in the revised version of the paper.

---

### Official Review · Reviewer_xoAh · 2024-07-12

**Soundness:** 3
**Presentation:** 3
**Contribution:** 2
**Rating:** 5
**Confidence:** 3

**Summary:**

In this paper, the authors present a comprehensive theoretical analysis using the contextual stochastic block model (CSBM) to evaluate the performance of vanilla graph convolution after removing the principal eigenvector to prevent over-smoothing. They conduct a spectral analysis for k rounds of corrected graph convolutions and provide findings for both partial and exact classification.

**Strengths:**

1. The paper provides both detailed theoretical analysis and experiments on three datasets: CORA, CiteSeer, and Pubmed.

2. The paper provides a novel insight on why corrected convolution can mitigate over-smoothing and improve classification accuracy.

**Weaknesses:**

1. The synthesized data by CSBM might be adequate to illustrate the binary classification case, but the multi-class case could be more complicated. For instance, it is illustrated in [1] that the effects of signed propagation under binary-class case and multi-class case could be quite different. It is mentioned that the authors would like to analyze the multi-class CSBM using more sophisticated architectures and I look forward to the further analysis.

2. Instead of stacking more than 20 layers of GNNs, it is a common practice to limit the number of layers to 3 to 5. According to Figure 3, the accuracy of the three learning methods appears more complex when the number of layers is limited to 5 or fewer. Therefore, the conclusion may not be applicable to real-world data applications.

[1] Choi, Yoonhyuk, et al. "Improving Signed Propagation for Graph Neural Networks." arXiv preprint arXiv:2301.08918 (2023).

**Questions:**

1. (Refer to weakness # 2) Would it be possible to analyze GNNs employing various convolution methods with shallower layers?

**Limitations:**

The authors have addressed some of the limitations of their work (limited to the binary classification case). Their work does not present any potential negative societal impact.

---

> ### Author Rebuttal · Authors · 2024-08-07
>
> We are grateful for the thorough feedback on our work. It certainly helped us improve our manuscript.
>
> > The multi-class case could be more complicated. It is mentioned that the authors would like to analyze the multi-class CSBM using more sophisticated architectures and I look forward to further analysis.
>
> We have extended our partial recovery result (Theorem 4.1) to the multi-class setting. Statement and proof of our result are in the global response. In our analysis, we have $L$ classes, and features are generated by a Gaussian mixture model with one mean for each class. We show that the graph signal is measured by the quantity $\lambda = \frac{(p-q)n}{dL}$ where $d$ is the expected degree. The graph noise is bounded by $\delta = O(\frac{1}{d}(\sqrt{np(1-p)/L} + \sqrt{nq(1-q)}))$. We assume that $|\lambda| \gtrsim k\delta$, and the cluster means are well separated. We show that after $k$ rounds of convolution of the data for sufficiently large $k$, $1-o(1)$ fraction of points will be closer to the means of their class than that of any other class. This means a classifier exists that correctly classifies $1-o(1)$ fraction of the data. Our bounds are similar to Theorem 4.1 in our paper. We refer the reveiwer to table 1 in the response pdf for detailed discussion of our theoretical guarantee in different regimes of parameters. We also provide experimental results on synthetic data for the multi-class CSBM that mirror our theoretical bounds (see the response pdf).
>
>
> Finally, we note that for multi-class CSBM, there is a wide range of assumptions one can make about the class sizes, edge probabilities, and distribution of features. One key assumption we made is that each class has one feature mean. Other models (ref line 385) have analyzed instances where one class can have multiple means. We believe we can extend our analysis to even broader multi-class settings in the future with additional techniques.
>
> > It is illustrated in [1] that the effects of signed propagation under the binary-class case and the multi-class case could be quite different.
>
> This is a good comment. We will discuss this in our revision along with the reference you provided. Indeed, the heterophilic setting in the multi-class problem can be more difficult than the homophilic setting. Using our new theorem above, we formalize this observation below using CSBM. Just as in the case of 2-classes, there is no difference in our analysis between the homophilic $(p>q)$ and heterophilic $(q > p)$ cases.
>
> Note that in the homophilic case, $\lambda>0$ and in the heterophilic case, $\lambda<0$. Implicitly, we are applying signed propagation in the heterophilic in Theorem 1 because of our scaling factor of $1/\lambda^k$. Thus, one can view it as applying the convolution $\tilde{A}/\lambda$ each time.
>
> We show that as long as $|\lambda| \gtrsim k\delta$, the data will become well-separated after $k$ convolutions. This is because $\tilde{A}^k$ will tend towards its dominant eigenspaces, which correspond to the eigenvalues that are large in absolute value. However, the value of $\delta$ itself can be much worse in the heterophilic case. In Lemma 2, our upper bound on the noise from the graph, $\delta$, scales with $\sqrt{np/L} + \sqrt{nq}$ when $p,q < 1/2$. Assuming we fix $\min(p,q)$ to be small, if $p >> q$, this quantity is much smaller than when $q >> p$. This suggests that while the signal strength from the graph is the same, the noise can become much worse if we have many classes in the heterophilic case.
>
>
> > Comment 3 and Question: It is a common practice to limit the number of layers to 3 to 5. According to Figure 3, the accuracy of the three learning methods appears more complex when the number of layers is limited to 5 or fewer. Therefore, the conclusion may not apply to real-world data applications. Would it be possible to analyze GNNs employing various convolution methods with shallower layers?
>
> We agree with the reviewer that for shallow networks, there might not be a performance difference between the three convolutions tested in our real experiments. Since the reviewer seems to ask for a theoretical comparison among the convolutions in our paper, we can compare our exact classification result (Theorem 4.2) with previous work in the same model but with uncorrected convolutions. We observe that even in the case of one convolution, under appropriate assumptions in CSBM, the uncorrected convolution achieves the same performance as the corrected convolution, thus verifying the reviewer's claim in synthetic data as well. In particular, in our Theorem 4.2, if $k = 1$ (one convolution) and $\frac{p-q}{p+q} = \Omega(1)$, then the exact classification threshold for the feature signal-to-noise ratio $\frac{\|\mu-\nu\|}{\sigma}$ is about $\sqrt{\frac{\log{n}}{np}}$. Theorem 1.2 of the reference in line 381 shows the same separability threshold for uncorrected convolutions in the same setting up to a factor of $\sqrt{\log{n}}$, which can be removed by more careful analysis (see ref line 385).
>
> In our paper, we quantify how many convolutions are necessary to attain optimal behavior in CSBM. In Section 4, we provide several examples of model parameter values for which only a constant number of convolutions is optimal. We provide more such examples in the multi-class model in our response pdf.
>
> Concerning a high number of convolutions, we note that oversmoothing is a well-studied problem in the GNN community, which we do not claim to have resolved. However, we believe our results offer new theoretical insights into the oversmoothing phenomenon that we hope could be useful in future studies constructing deeper GNNs that are more effective.
>
> Final remarks
>
> If the reviewer has more actionable suggestions to further improve our work, please kindly let us know. We would be happy to address any further questions. We hope the clarifications are sufficient for them to consider raising their score.

---

> > ### Comment · Reviewer_xoAh · 2024-08-09
> > **Thank you for the response**
> >
> > The authors' response has adequately addressed my main concerns on the multi-class scenario and the analysis on shallower networks. I have raised my rating.

---

> > > ### Author Response · Authors · 2024-08-10
> > >
> > > We thank the reviewer for raising their score. We will add the additional result in the revised version of the paper and address all comments.

---

### Author Rebuttal · Authors · 2024-08-07

$$
   \def\norm#1{{\|#1\|}}
   \def\E{{\mathbb{E}}}
$$

## Multi-class Analysis
We define the L-Block CSBM with parameters $p,q,L,n,m$. We have $n$ nodes and $L$ classes, $\mathcal{C}_1,...\mathcal{C}_L$, of size $n/L$. For each node $i$, we have a feature vector $x_i\in \mathbb{R}^m$ with distribution $N(\mu_i, \sigma^2 I_m)$. For each class $\mathcal{C}_l$, $l\in [L]$, we let $c_l \in \mathbb{R}^m$ be the class mean, and for $i\in \mathcal{C}_l$, $\mu_i = c_l$. We define

$$\tilde{A} = \frac{1}{d}A - \frac1n\mathbf{1}\mathbf{1}^\top,$$
where $d = \frac{np}{L} + \frac{nq(L-1)}{L}$ is the expected degree of each vertex. Note $d$ can be estimated accurately with high probability by degree concentration (see proposition 5.2). We introduce useful notation below.

- **Graph Signal:** $\lambda := \frac{(p-q)n}{dL}$ is the strength of the signal from the graph.
- **Graph Noise:** $\delta := C(\frac{1}{d}(\sqrt{np(1-p)/L} +\sqrt{nq(1-q)}))$ for some constant $C$. $\delta$ is an upper bound on the graph noise.
- Let $U:=\E[X]$ be the matrix whose $i^{th}$ row is $\mu_i$. We will use $\norm{U}_F^2$ to denote its Frobenius norm. We also assume our features are centered on expectation so that $U^\top \mathbf{1} = 0$. This is not restrictive since it can be satisfied by applying a linear shift to the features.
- Let $\Delta = \min_{i,j\in [n]}\norm{\mu_i-\mu_j}$ be the minimum distance between the centers.

### Theorem 1

Given the CSBM with parameters, $p,q,L,n,m$, suppose $\min(p,q) \geq \Omega(\frac{\log^{2}n}{n})$ and $|\lambda| > 4k\delta$. Let $X^{(k)} = \frac{1}{\lambda^k}A^{(k)}X$ be the feature matrix after $k$ rounds of convolutions with scalaing factor $1/\lambda^k$. Let $x^{(k)}_i$ be the $i^{th}$ row of the matrix $X^{(k)}$. Then with probability $1-n^{-\Omega(1)}$, at least $n - n_e$ nodes, $i$, satisfy $\norm{x^{(k)}_i - \mu_i}< \Delta/2$ where
$$n_e =  O\Big((k \delta/|\lambda|)^2\frac{\norm{U}_F^2}{\Delta^2} + (L + n(\delta/|\lambda|)^{2k})\frac{\sigma^2m\log{n}}{\Delta^2}\Big).$$

In particular, the quadratic classifer $x\mapsto \text{softmax}(\norm{x-c_l}^2)_{l=1}^L$ will correctly classify at least $n-n_e$ points.

---

See the attached PDF for examples of our bound in specific setting as well as experimental results.

The following lemma explains how the expressions for graph signal and noise are derived.

### Lemma 2
The convolution matrix $\tilde{A}$ can be decomposed as $\tilde{A} = M + R'$ where:

- $M = \E[\tilde{A}]$ has rank $L-1$, with $L-1$ eigenvalues equalled to $\frac{(p-q)n}{dL} = \lambda$. Also, $MU = \lambda U$
- $R'$ is a random matrix such that with probability $\ge 1-n^{-\Omega(1)}$, $\norm{R'} \leq O(\frac{1}{d}(\sqrt{np(1-p)/L} + \sqrt{nq(1-q)})) = \delta$

---

The proof is standard, so we only give a sketch due to space limitations. We are happy to give the full proof in the discussion period. For item 1, we note $\E[A]$ is rank $L$ and has top eigenvector $\mathbf{1}$ with eigenvalue $d$. Its second eigenvalue is $\lambda$ with multiplicity $L-1$ and the eigenspace is characterized by the set vectors orthogonal to $\mathbf{1}$ and constant on each class. Since each class has one center and $U^\top \mathbf{1} = 0$, the columns of $U$ are in the second eigenspace. To obtain item 2, we decompose $A-E[A]$ into entries with probability $p$ and entries with probability $q$. Then we bound them separately using matrix concentration in theorem A.4 of the paper.

We will also need to bound the operator norm distance between the $k^{th}$ convolution and $M^k$
### Lemma 3

Suppose $|\lambda| > 4k\delta$. Then with high probability, we have
$$\norm{\frac{1}{\lambda^k}(\tilde{A}^k - M^k)} \leq 2k\delta/|\lambda|.$$

#### Proof
By Lemma 2, we have
$$\tilde{A}^k = (M+R')^k = M^k + \sum_{l=1}^k\sum_{b\in {[k]\choose l}}\prod_{i=1}^k M^{1-b(i)}R'^{b(i)},$$
where the inner sum is over bit-strings $b$ of length $k$ with exactly $l$ $1$'s and $k-l$ $0$'s. Note that $\norm{M} = |\lambda|$ and $\norm{R'}\leq \delta$ with high probability. Using the fact that $\norm{AB}\leq \norm{A}\norm{B}$ and triangle inequality, we have
$$\norm{\frac{1}{\lambda^k}((M+R')^k - M^k)} \leq \frac{1}{|\lambda|^k}\sum_{l=1}^k{k\choose l}\norm{M}^{k-l}\norm{R'}^{l} \leq \sum_{l=1}^k{k\choose l}(\frac{\delta}{|\lambda|})^{l} = (1+\frac{\delta}{|\lambda|})^{k}-1$$
Our assumption that $|\lambda| > 4\delta k$ implies the RHS is at most $2k\delta/|\lambda|$.

---

Now we are ready to prove our main theorem.
#### Proof of Theorem 1
We decompose our data as $X = U + G$, where $G$ is a Gaussian matrix with i.i.d $N(0, \sigma^2)$ entries. We will decompose our error into error from the graph and error from the feature noise. Recall that we take our scaling factor to be $1/\lambda^k$. We have
$$\norm{X^{(k)}-U}_F^2 = \norm{\frac{1}{\lambda^k}\tilde{A}^k(U + G)-U}_F^2 \leq 2\norm{\frac{1}{\lambda^k}\tilde{A}^kU - U}_F^2 + 2\norm{\frac{1}{\lambda^k}\tilde{A}^k G}_F^2.$$

By Lemma 2, we have $MU = \lambda U$. Thus, we have
$$\norm{\frac{1}{\lambda^k}\tilde{A}^kU - U}_F^2 = \norm{\frac{1}{\lambda^k}(\tilde{A}^k - M^k)U}_F^2 \leq \norm{\frac{1}{\lambda^k}(\tilde{A}^k - M^k)}^2\norm{U}_F^2 \leq (k\delta/|\lambda|)^2\norm{U}_F^2,$$
where the inequality follows from Lemma 3. By standard Gaussian norm concentration for $G$ (see theorem A.1 of paper), we have high probability
$$\norm{\frac{1}{\lambda^k}\tilde{A}^k G}_F^2 \leq \frac{1}{\lambda^{2k}}Tr(\tilde{A}^{2k})\sigma^2m\log{n}$$
Since adding $R'$ to $M$ preturbs its eigenvalues by at most $\delta$ (see theorem A.1 in the paper), we have
$$\frac{1}{\lambda^{2k}}Tr(\tilde{A}^{2k}) \leq (1 + \delta/|\lambda|)^{2k}(L-1) + n(\delta/|\lambda|)^{2k}$$

Note that $|\lambda| > 4\delta k$ implies $(1 + \delta/|\lambda|)^{2k} \leq 1$. Now let $n_e$ be the number of points $i$ such that $\norm{x^{(k)}_i - \mu_i} \geq \Delta/2$. Then we have
$$n_e\Delta^2\leq  4\norm{X^{(k)}-U}_F^2 \leq O((k\delta/|\lambda|)^2\norm{U}_F^2 + (L + n(\delta/|\lambda|)^{2k})m\sigma^2\log{n}).$$

---

### Decision · Program_Chairs · 2024-09-25

**Decision:**

Accept (poster)

**Comment:**

This paper provides a spectral analysis showing that, for data generated by stochastic block models, the oversmoothing phenomenon can be alleviated by subtracting the projection upon the leading eigenvector from the shift operator. The new leading eigenvector then becomes the second eigenvector, which contains more information about the structure of the graph than the original leading eigenvector, related to properties of connectedness of the graph, i.e. information about sparse bipartitions. This idea leads to a practical approach for correcting the shift operator in standard GNNs, and to theoretical guarantees on node classification on 2-class stochastic block models. The improvement in classification is demonstrated on synthetic data and on citation networks, where GCN with the corrected shift operator outperforms vanilla GCN. While the experimental results are limited, they are sufficient for a paper with a theoretical contribution. Overall, this is a solid theoretical work with sufficient novelty.

The authors are asked to incorporate the reviewer comments for the camera ready version.